# KALE Flow: A Relaxed KL Gradient Flow for Probabilities with Disjoint Support

**Pierre Glaser**
Gatsby Computational Neuroscience Unit
University College London
pierreglaser@gmail.com

**Michael Arbel**
Université Grenoble Alpes, Inria, CNRS,
Grenoble INP, LJK,38000 Grenoble, France*
michael.n.arbel@gmail.com

**Arthur Gretton**
Gatsby Computational Neuroscience Unit
University College London
arthur.gretton@gmail.com

## Abstract

We study the gradient flow for a relaxed approximation to the Kullback-Leibler (KL) divergence between a moving source and a fixed target distribution. This approximation, termed the KALE (KL Approximate Lower bound Estimator), solves a regularized version of the Fenchel dual problem defining the KL over a restricted class of functions. When using a Reproducing Kernel Hilbert Space (RKHS) to define the function class, we show that the KALE continuously interpolates between the KL and the Maximum Mean Discrepancy (MMD). Like the MMD and other Integral Probability Metrics, the KALE remains well-defined for mutually singular distributions. Nonetheless, the KALE inherits from the limiting KL a greater sensitivity to mismatch in the support of the distributions, compared with the MMD. These two properties make the KALE gradient flow particularly well suited when the target distribution is supported on a low-dimensional manifold. Under an assumption of sufficient smoothness of the trajectories, we show the global convergence of the KALE flow. We propose a particle implementation of the flow given initial samples from the source and the target distribution, which we use to empirically confirm the KALE's properties.

## 1 Introduction

We consider the problem of transporting probability mass from a source distribution $\mathbb{P}$ to a target distribution $\mathbb{Q}$ using a Wasserstein gradient flow in probability space. When the density of the target is well-defined and available, the Wasserstein gradient flow of the Kullback-Leibler (KL) divergence provides a simple way to transport mass towards the target through the Fokker-Planck equation as established in the seminal work of [31]. Its time discretization yields a practical algorithm, the Unadjusted Langevin Algorithm (ULA), which comes with strong convergence guarantees [22, 19]. A more recent gradient flow approach, Stein Variational Gradient Descent (SVGD) [36], also leverages the analytic expression of the density and constructs a gradient flow of the KL, albeit using a metric different from the Wasserstein metric.

The KL divergence is of particular interest due to its information theoretical interpretation [56] and its use in Bayesian Inference [13]. The KL defines a strong notion of convergence between probability distributions, and as such is often widely used for learning generative models, through Maximum Likelihood Estimation [20]. Using the KL as a loss requires knowledge of the density of the target, however; moreover, this loss is well-defined only when the distributions share the same support. Consequently, we cannot use the KL in settings where the probability distributions are mutually singular, or when they are only accessible through samples. In particular, the Wasserstein gradient flow of the KL in these settings is ill-defined.

---

*Work mostly completed at the Gatsby Unit.

35th Conference on Neural Information Processing Systems (NeurIPS 2021).

Recent works have considered the gradient flow of Integral Probability Metrics (IPM) [46] instead of the KL, in settings where only samples (and not the density) of the target are known. This includes the Maximum Mean Discrepancy (MMD) [4] and the Kernelized Sobolev Discrepancy (KSD) [45, 44]. One motivation for considering these particle flows is their connection with the training of Generative Adversarial Networks (GANs) [28] using IPMs such as the Wasserstein distance [7, 29, 27], the MMD [24, 34, 33, 9, 10, 5] or the Sobolev discrepancy [43]. As discussed in [45, Section 3.3], these flows define update equations that are similar to those of a generator in a GAN. Thus, studying the convergence flows can provide helpful insight into conditions for GAN convergence, and ultimately, improvements to GAN training algorithms. A second motivation lies in the connection between the training dynamics of infinitely wide 2-layer neural networks and the Wasserstein gradient flow of particular functionals [52]. Thus, analyzing the asymptotic behavior of such flows [40, 58, 18] can ultimately provide convergence guarantees for the training dynamics of neural networks. Establishing such results remains challenging for some classes of IPMs, however, such as the MMD [4].

In this paper, we construct the gradient flow of a relaxed approximation of the KL, termed the KALE (KL Approximate Lower bound Estimator). Unlike the KL, the KALE is well-defined given any source and target, regardless of their relative absolute continuity. The KALE is obtained by solving a regularized version of the Fenchel dual problem defining the KL, defined over a restricted function class [48, 6], and can be estimated solely from samples from the data. The version of the KALE we consider in this work benefits from two important features that are crucial for defining and analyzing a *relaxed* gradient flow of the KL. (1) We define the function class to be a Reproducing Kernel Hilbert Space (RKHS). This makes the optimization problem defining the KALE convex and allows for practical algorithms computing it. (2) We consider a regularized version of the problem defining the KALE, thus providing a simpler expression for the gradient flow by virtue of the envelope theorem [42]. In Section 2, we review the KALE, and show that it is a divergence that metrizes the weak convergence of probability measures, while interpolating between the KL and the MMD depending on the amount of regularization. We then construct in Section 3 the Wasserstein Gradient Flow of the KALE, and we show global convergence of the KALE flow provided that the trajectories are sufficiently regular. In Section 4, we introduce the *KALE particle descent* algorithm as well as a practical way to implement it. In Section 5, we present the results obtained by running the KALE particle descent algorithm on a set of problems with different geometrical properties. We show empirically that the sensitivity to support mismatch of the KALE inherited from the KL leads to well-behaved trajectories compared to the MMD flow, making the KALE flow a desirable alternative when a KL flow cannot be defined.

**Related work.** The Fenchel dual formulation of the KL, and more generally $f$-divergences, has a rich history in Machine Learning: [48] relied on this dual formulation to estimate the KL between two probability distributions when their density ratios belong to an approximating class. They derived a plug-in estimator for the KL which comes with convergence guarantees. In the context of GANs, [49] used the Fenchel dual representation of $f$-divergences, of which the KL is a particular instance, as a GAN critic. Later, [41] used Fenchel duality to estimate the KL in the context of Variational Inference (VI) when the variational distribution is chosen to be an implicit model, thus allowing more flexible models at the expense of tractability of a KL term appearing in the expression of the ELBO. In both the GAN and VI settings, the function class defining the $f$-divergence was restricted to neural networks. Recently, [6] showed that controlling the smoothness of such a function class results in a divergence, the KL Approximate Lower bound Estimator (KALE), that metrizes the *weak convergence of distributions* [21], unlike the KL which defines a stronger topology [62]. The KALE is therefore well-suited for learning Implicit Generative Models which are only accessible through sampling, as advocated in [8]. When neural network classes are used, however, the method has no optimization guarantees, as the dual problem becomes non-convex due to the choice of the function class. This is unlike our setting [and that of 48], since our dual problem is strongly convex and comes with guarantees. [12] presented a general framework incorporating both $f$-divergences and Integral Probability Metrics, but rely on another variational formulation than the Fenchel dual one. In parallel to work related to $f$-divergences, [30, 16, 53, 1] have investigated the task of sampling in the case where the source and the target have disjoint supports. Again, unlike our setting, these works assume that the log-density of the target distribution is known.

## 2 Interpolating between KL and MMD using KALE

In this section, we introduce the *KALE*, a relaxed approximation of the KL divergence. Although we will use the KALE to define a relaxed KL gradient flow, we show in this section that the KALE is an object of independent interest outside the gradient flow setting: indeed, it is a valid *probability divergence* that metrizes the weak convergence of probability distributions, and interpolates between the KL and the Maximum Mean Discrepancy.

**Mathematical details and notation**  We start by introducing some notation. We denote by $\mathcal{P}(\mathbb{R}^d)$ the set of probability measures defined on $\mathbb{R}^d$ endowed with its Borelian $\sigma$-algebra, and by $\mathcal{P}_2(\mathbb{R}^d) \subset \mathcal{P}(\mathbb{R}^d)$ the set of elements of $\mathcal{P}(\mathbb{R}^d)$ with finite second moment. Weak convergence of a sequence of probability measures $(\mathbb{P}_n)_{n \geq 0}$ towards $\mathbb{P}$ is written $\mathbb{P}_n \rightharpoonup \mathbb{P}$. A positive definite kernel on the set $\mathbb{R}^d$ will be denoted $k : \mathbb{R}^d \times \mathbb{R}^d \longmapsto \mathbb{R}$, with RKHS $\mathcal{H}$, and we will use $k_x$ to refer to the RKHS function $y \longmapsto k(x, y)$ obtained by fixing the first argument of $k$. The Dirac delta measure for $x \in \mathbb{R}^d$ will be written $\delta_x$. We denote by $C_c^\infty(\mathbb{R}^d \times (0, +\infty))$ the set of infinitely differentiable functions with compact support on $\mathbb{R}^d \times (0, +\infty)$, and by $C_b^0(\mathbb{R}^d)$ the set of continuous bounded functions from $\mathbb{R}^d$ to $\mathbb{R}$. Sets of $N$ points in $\mathbb{R}^d$ will be indexed using a superscript $\{x^{(i)}\}_{i=1}^N$, while a sequence of points in $\mathbb{R}^d$ will use a subscript: $(x_n)_{n \in \mathbb{N}}$. If random, elements of such sets $X^{(i)}$ or iterates of such sequences $X_n$ will be capitalized. If not, they will be kept in lower-case. For the sake of notational lightness, the choice of the norm used for a specific object (vectors, functions, operators) will be specified with a subscript (e.g. $\|h\|_{\mathcal{H}}$ for the RKHS norm) only if the said choice is not obvious from the context. This remark also holds when referring to the null element of a vector space ($0_{\mathcal{H}}, 0_{\mathbb{R}^d}, ...$).

### 2.1 The KL Approximate Lower bound Estimator (KALE)

The central equation to derive the KALE is the (Fenchel) dual formulation of the KL [3, Lemma 9.4.4]:

$$\text{KL}(\mathbb{P} \,||\, \mathbb{Q}) = \sup_{h \in C_b^0(\mathbb{R}^d)} \left\{ 1 + \int h d\mathbb{P} - \int e^h d\mathbb{Q} \right\}. \tag{1}$$

KALE is obtained from Eq. (1) by restricting the variational set to an RKHS $\mathcal{H}$ with reproducing kernel $k$, and by adding a penalty to the objective that controls the RKHS norm of the test function $h$. This regularization ensures that the KALE is well-defined for a broader class of probabilities compared to the KL, even when $\mathbb{P}$ and $\mathbb{Q}$ are mutually singular. Its complete definition is stated below:

**Definition 1** (KALE). *Let $\lambda > 0$, and $\mathcal{H}$ be an RKHS with kernel $k$. The Kullback-Leibler Approximate Lower bound Estimator (KALE) is given by:*

$$KALE(\mathbb{P} \,||\, \mathbb{Q}) = (1 + \lambda) \max_{h \in \mathcal{H}} \left\{ 1 + \int h d\mathbb{P} - \int e^h d\mathbb{Q} - \frac{\lambda}{2} \|h\|_{\mathcal{H}}^2 \right\}. \tag{2}$$

The $(1 + \lambda)$ scaling will prevent a degenerate decay to 0 in the large $\lambda$ regime (see Proposition 1). The definition we consider here also differs from the one in [6], which first finds the optimal function $h^\star$ solving Eq. (2), and then defines KALE by evaluating the KL objective in Eq. (1), thereby discarding the regularization term when evaluating the divergence.

**Mathematical Assumptions**  To prove the theoretical results stated in this work, we will make the following basic assumptions on the kernel $k$:

**Assumption 1** (Boundedness). *There exists $K > 0$ such that $k(x, x) \leq K$, for all $x \in \mathbb{R}^d$.*

**Assumption 2** (Smoothness). *The kernel is $2$-times differentiable in the sense of [60, Definition 4.35]: for all $i, j \in \{1, \ldots, d\}$ $\partial_i \partial_{i+d} k$ and $\partial_i \partial_j \partial_{i+d} \partial_{j+d} k$ exist. Moreover, we have: $\|\nabla_1 k_x\|^2 \triangleq \sum_{i=1}^d \|\partial_i k_x\|^2 \leq K_{1d}$ and $\|\mathbf{H}_1 k_x\|^2 \triangleq \sum_{i,j=1}^d \|\partial_i \partial_j k_x\|^2 \leq K_{2d}$, where $d$ indicates an expected scaling with dimension.*

Assumption 1 guarantees the integrability of the objects intervening in KALE, and implies boundedness of the RKHS functions. Assumption 2 guarantees first and second order smoothness of the RKHS functions, a property invoked to control the KALE flow trajectories. Indeed, both the

differential and the hessian of any $f \in \mathcal{H}$ can now be bounded in operator norm: using the Cauchy-Schwarz inequality and the kernel reproducing derivative property [60, Corollary 4.36], we have: $|\partial_i f(x)| \leq \|\partial_i k_x\| \|f\|$ and $|\partial_i \partial_j f(x)| \leq \|\partial_i \partial_j k_x\| \|f\|$, implying $\|\nabla f(x)\| \leq \sqrt{K_{1d}} \|f\|$, and $\|\boldsymbol{H}(f(x))\|_{\mathrm{Op}} \leq \|\boldsymbol{H}(f(x))\|_{\mathrm{F}} \leq \sqrt{K_{2d}} \|f\|$.

**KALE is a probability divergence**  We first show that KALE is a probability divergence, and presents topological properties compatible with its use in generative models, such as GANs and Adversarial VAEs: weak continuity, and metrizing the weak convergence of probability distributions. We recall that a functional $\mathcal{D}(\cdot \| \cdot)$ is a probability divergence if both $\mathcal{D}(\mathbb{P} \| \mathbb{Q}) \geq 0$ and $\mathcal{D}(\mathbb{P} \| \mathbb{Q}) = 0 \iff \mathbb{P} = \mathbb{Q}$, for any $\mathbb{P}, \mathbb{Q} \in \mathcal{P}(\mathbb{R}^d)$.

**Theorem 1** (Topological properties of KALE). *Let $\mathbb{P}, \mathbb{Q} \in \mathcal{P}(\mathbb{R}^d)$. Let $(\mathbb{P}_n)_{n \geq 0}$ be a sequence of probability measures. Then, under Assumption 1:*

*(i) KALE is weakly continuous: $\mathbb{P}_n \rightharpoonup \mathbb{P} \implies \lim_{n \to \infty} KALE(\mathbb{P}_n \| \mathbb{Q}) = KALE(\mathbb{P} \| \mathbb{Q})$*

*(ii) If $k$ is universal [57], then for any $\lambda > 0$, KALE is a probability divergence. Moreover, KALE metrizes the weak topology between probability measures with finite first order moments.*

Central to the proof of all points in this theorem is a link between KALE and the MMD witness function $f_{\mathbb{P},\mathbb{Q}}$, which we report in the next lemma. We recall that given an RKHS $\mathcal{H}$ associated to a kernel $k$, and two probability distributions $\mathbb{P}$ and $\mathbb{Q}$, the MMD is defined as the RKHS norm of the difference of mean embeddings of $\mathbb{P}$ and $\mathbb{Q}$:

$$\mathrm{MMD}(\mathbb{P} \| \mathbb{Q}) = \|f_{\mathbb{P},\mathbb{Q}}\| \quad (f_{\mathbb{P},\mathbb{Q}} = \int k_x \mathrm{d}\mathbb{P}(x) - \int k_x \mathrm{d}\mathbb{Q}(x) \triangleq \mu_{\mathbb{P}} - \mu_{\mathbb{Q}}). \tag{3}$$

**Lemma 1.** *Let $\mathbb{P}, \mathbb{Q} \in \mathcal{P}(\mathbb{R}^d)$, and $\mathcal{K} : \mathcal{H} \longmapsto \mathbb{R}$ be the objective maximized by KALE, e.g. $\mathcal{K}(h) = 1 + \int h \mathrm{d}\mathbb{P} - \int e^h \mathrm{d}\mathbb{Q} - \frac{\lambda}{2} \|h\|^2$. Then, under Assumption 1, $\mathcal{K}$ is Fréchet differentiable. Moreover, the following relationship holds:*

$$\nabla \mathcal{K}(0) = f_{\mathbb{P},\mathbb{Q}}$$

Intuitively, noting that $\mathcal{K}(0) = 0$, Lemma 1 ensures that KALE presents "equivalent" regularity and discriminative properties to those of MMD (a divergence which is itself, under the assumptions of this theorem, weakly continuous and that metrizes the weak convergence of probability distributions). The proof of the second point of Theorem 1 is inspired by [6], which in turn derives from [66, 37], and is adapted to account for the extra norm penalty term in the version of the KALE in this paper.

**Interpolating between the MMD and the KL using the KALE**  The KALE includes a positive regularization parameter $\lambda$, inducing two asymptotic regimes: $\lambda \to 0$ and $\lambda \to \infty$. In these regimes, the KALE asymptotically recovers on the one hand the KL divergence, and on the other hand the MMD.

**Proposition 1** (Asymptotic properties of KALE). *Let $\mathbb{P}, \mathbb{Q} \in \mathcal{P}(\mathbb{R}^d)$. Then, under Assumption 1, the following result holds:*

$$\lim_{\lambda \to +\infty} KALE(\mathbb{P} \| \mathbb{Q}) = \frac{1}{2} MMD^2(\mathbb{P} \| \mathbb{Q}). \tag{4}$$

*Suppose additionally that $\log \frac{d\mathbb{P}}{d\mathbb{Q}} \in \mathcal{H}$. Then,*

$$\lim_{\lambda \to 0} KALE(\mathbb{P} \| \mathbb{Q}) = KL(\mathbb{P} \| \mathbb{Q}). \tag{5}$$

Proposition 1 shows that the MMD can be seen as solving a degenerate version of the KL objective. Eq. (5) is natural given the original definition of the KALE, and highlights the continuity of the KALE objective w.r.t the regularization parameter $\lambda$. Both the MMD and the KL exhibit limitations when used for defining gradient flows, however: as discussed in [4, 25, 14], the MMD induces a "flat" geometry, making its use in generative models tricky [5]. On the other hand, the KL comes with stronger convergence guarantees [3], but its use in sampling algorithms is limited to cases where the target distribution has a density, discarding cases satisfying the widely known *manifold hypothesis* [47, 14, 17], stating that typical high dimensional data used in machine learning are distributed on a lower-dimensional manifold. For this reason, we argue that the true interest of the KALE does not lie in its interpolation properties, but rather in the geometry it generates at intermediate values of $\lambda$.

**The KALE's dual objective** Interestingly, the KALE itself admits a dual formulation, with a strong connection to the original KL expression:

$$\begin{cases} \text{KALE}(\mathbb{P} \, || \, \mathbb{Q}) \propto \min_{f>0} \left\{ \int (f(\log f - 1) + 1) \, \mathrm{d}\mathbb{Q} + \frac{1}{2\lambda} \left\| \int f(x) k_x \mathrm{d}\mathbb{Q}(x) - \mu_{\mathbb{P}} \right\|_{\mathcal{H}}^2 \right\} \\ h^\star = \int f^\star(x) k_x \mathrm{d}\mathbb{Q}(x) - \mu_{\mathbb{P}} \end{cases} \tag{6}$$

The solution $f^\star$ of Eq. (6) can be seen as an entropically-regularized density ratio estimate on the support of $\mathbb{Q}$ (additional details on the KALE dual objective are given in the appendix). Eq. (6) also yields an elegant estimation procedure, as discussed below.

**Computing KALE**$(\mathbb{P} \, || \, \mathbb{Q})$ **in practice** As for other IPMs, computing KALE$(\mathbb{P} \, || \, \mathbb{Q})$ for arbitrary $\mathbb{P}$ and $\mathbb{Q}$ is intractable, and is therefore approximated using a discretization procedure. A common procedure is to assume access to samples $\{Y^{(i)}\}_{i=1}^N$ and $\{X^{(i)}\}_{i=1}^N$ from $\mathbb{P}$ and $\mathbb{Q}$ and to solve the empirical equivalent of Eq. (6) (e.g. Eq. (6), but where $\mathbb{P}$ and $\mathbb{Q}$ are replaced by their plug-in estimators $\widehat{\mathbb{P}}^N = \frac{1}{N} \sum_{i=1}^N \delta_{Y^{(i)}}$ and $\widehat{\mathbb{Q}}^N = \frac{1}{N} \sum_{i=1}^N \delta_{X^{(i)}}$). This empirical equivalent is written

$$\min_{f>0} \quad \frac{1}{N} \sum_{i=1}^N f(X^{(i)}) \log(f(X^{(i)})) - f(X^{(i)}) + 1 + \frac{1}{2\lambda} \left\| \frac{1}{N} \sum_{i=1}^N f(X^{(i)}) k(X^{(i)}, \cdot) - \mu_{\widehat{\mathbb{P}}} \right\|_{\mathcal{H}}^2 \tag{7}$$

which is a strongly convex $N$-dimensional problem, and can be solved using standard euclidean optimization methods. By adapting arguments of [6], it can be shown that the discrepancy between the KALE's empirical and population value, $|\text{KALE}(\widehat{\mathbb{P}}^N \, || \, \widehat{\mathbb{Q}}^N) - \text{KALE}(\mathbb{P} \, || \, \mathbb{Q})|$ (often called "sample complexity"), is at most $O(\frac{1}{\sqrt{N}})$. This rate is identical that of Sinkhorn divergences [26], another family of entropically-regularized divergences.

## 3 KALE Gradient Flow

Having introduced KALE as a relaxed approximation of the KL, we now construct the KALE gradient flow, and assert its well-posedness. We provide conditions for global convergence of the flow, and discuss its relationship with the MMD flow and the KL flow. All proofs are given in the appendix.

### 3.1 Wasserstein Gradient Flow of the KALE

*Wasserstein Gradient Flows* of divergence functionals $\mathcal{F}(\mathbb{P} \, || \, \mathbb{Q})$ aim at transporting mass from an initial probability distribution $\mathbb{P}_0$ towards a target distribution $\mathbb{Q}$ by following a path $\mathbb{P}_t$ in probability space. The path is required to dissipate energy, meaning that $t \mapsto \mathcal{F}(\mathbb{P}_t \, || \, \mathbb{Q})$ is a decreasing function of time. Additionally, it is constrained to satisfy a continuity equation that allows only local movements of mass without jumping from a location to another. This equation involves a time dependent vector field $V_t$ which serves as a force that drives the movement of mass at any time $t$:

$$\partial_t \mathbb{P}_t + \text{div}(\mathbb{P}_t V_t) = 0. \tag{8}$$

Eq. (8) holds in the *sense of distributions*, meaning that for any test function $\varphi \in C_c^\infty(\mathbb{R}^d \times (0, +\infty))$, we have:

$$\int \partial_t \varphi(x, t) \mathrm{d}\mathbb{P}_t \mathrm{d}t + \int \langle \nabla_x \varphi(x, t), V_t \rangle_{\mathbb{R}^d} \, \mathrm{d}\mathbb{P}_t \mathrm{d}t = 0.$$

The Wasserstein gradient flow of a (sufficiently regular) functional $\mathcal{F}$ is then obtained by choosing $V_t$ as the gradient of *first variation* of $\mathcal{F}$, defined as the Gâteaux derivative of $\mathbb{P}$ along the direction $\chi$,

$$\mathcal{D}_{\mathbb{P}} \mathcal{F}(\mathbb{P}; \chi) \triangleq \lim_{\epsilon \to 0} \epsilon^{-1} \left( \mathcal{F}(\mathbb{P} + \epsilon \chi) - \mathcal{F}(\mathbb{P}) \right),$$

where $\int d\chi = 0$, and provided that such a limit exists. This choice recovers a particle *Euclidean gradient flow* when $\mathbb{P}_0$ is a finite sum of Dirac distributions, and can thus be seen as a natural extension of gradient flows to the space of probability distributions [3, 63, 64]. In the next proposition, we show that the functional $\mathbb{P} \longmapsto \text{KALE}(\mathbb{P} \, || \, \mathbb{Q})$ admits a well-defined gradient flow of this form.

**Proposition 2** (KALE Gradient Flow). *Let $\lambda > 0$, and $\mathbb{P}_0, \mathbb{Q} \in \mathcal{P}_2(\mathbb{R}^d)$. Under Assumptions 1 and 2, the Cauchy problem*

$$\partial_t \mathbb{P}_t - div(\mathbb{P}_t(1 + \lambda)\nabla h_t^\star) = 0, \quad \mathbb{P}_{t=0} = \mathbb{P}_0, \tag{9}$$

*where $h_t^\star$ is the unique solution of*

$$h_t^\star = \arg\max_{h \in \mathcal{H}} \left\{ 1 + \int h d\mathbb{P}_t - \int e^h d\mathbb{Q} - \frac{\lambda}{2} \|h\|^2 \right\}, \tag{10}$$

*admits a* unique *solution* $(\mathbb{P}_t)_{t \geq 0}$, *which is the* Wasserstein Gradient Flow *of the KALE.*

The proof, given in Appendix D, requires studying the regularity properties of KALE in the *metric space* $\mathcal{P}_2(\mathbb{R}^d)$ endowed with the Wasserstein-2 Distance [64], using the framework developed in [3].

### 3.2 Convergence properties of the KALE flow

Proposition 2 hints at a connection between the KALE flow and the MMD flow, which solves:

$$\partial_t \mathbb{P}_t - \text{div}\left(\mathbb{P}_t \nabla f_{\mathbb{P}_t, \mathbb{Q}}\right) = 0, \quad \mathbb{P}_{t=0} = \mathbb{P}_0 \tag{11}$$

The MMD flow and the KALE flow thus differ in the choice of witness function characterizing their velocity field. A convergence analysis of the MMD flow was proposed for a wide range of kernels in [4] using inequalities of Lojasiewicz type; in particular, the MMD flow is guaranteed to converge provided that the quantity $\mathbb{P}_t - \mathbb{Q}$ remains bounded in the *negative Sobolev distance* $\|\mathbb{P}_t - \mathbb{Q}\|_{\dot{H}^{-1}(\mathbb{P}_t)}$ [50]. We recall that the negative *weighted negative Sobolev distance* [4] between $\mu$ and $\nu$ is defined as:

$$\|\mu - \nu\|_{\dot{H}^{-1}(\mathbb{P})} = \sup_{\|f\|_{\dot{H}(\mathbb{P})} \leq 1} \left| \int f d(\mu - \nu) \right|,$$

which is obtained by duality with the weighted Sobolev semi-norm $\|f\|_{\dot{H}(\mathbb{P})} = (\int \|\nabla f\|^2 d\mathbb{P})^{\frac{1}{2}}$. Note the important role of the latter quantity in the energy dissipation formula of the KALE gradient flow:

$$\frac{d\text{KALE}(\mathbb{P}_t \,||\, \mathbb{Q})}{dt} = -\int (1 + \lambda)^2 \|\nabla h^\star\|^2 d\mathbb{P} = -(1 + \lambda)^2 \|h^\star\|_{\dot{H}(\mathbb{P})}^2. \tag{12}$$

In the next proposition, we extend the condition ensuring the global convergence of the MMD flow [4] to the KALE flow:

**Proposition 3.** *Under Assumptions 1 and 2, if $\|\mathbb{P}_t - \mathbb{Q}\|_{\dot{H}^{-1}(\mathbb{P}_t)} \leq C$ for some $C > 0$, then:*

$$KALE(\mathbb{P}_t \,||\, \mathbb{Q}) \leq \frac{C}{CKALE(\mathbb{P}_0 \,||\, \mathbb{Q}) + t}.$$

Proposition 3 ensures a convergence rate in $\mathcal{O}(1/t)$ provided that $\|\mathbb{P}_t - \mathbb{Q}\|_{\dot{H}^{-1}(\mathbb{P}_t)}$ remains bounded. This convergence rate is slower than the linear rate of the KL along its gradient flow [38] and could be an effect of RKHS smoothing.

## 4 KALE Particle Descent

We now derive a practical algorithm that computes the solution of a KALE gradient flow, given an initial source-target pair $\mathbb{P}_0$ and $\mathbb{Q}$. Because of the continuous-time dynamics, and the possibly continuous nature of $\mathbb{P}_0$ and $\mathbb{Q}$, solutions of Eq. (9) are intractable to compute and manipulate. To address this issue, we first introduce the *KALE Particle Descent Algorithm* that returns a sequence $(\widehat{\mathbb{P}}_n^N)_{n \geq 0}$ of discrete probability measures able to approximate the forward Euler discretization of $\mathbb{P}_t$ with arbitrary precision. Additionally, we show that the KALE particle descent algorithm can be regularized using *noise injection* [4], which guarantees global convergence of the flow under a suitable noise schedule. All proofs are given in the appendix.

## 4.1 The KALE Particle Descent Algorithm

**Time-discretized KALE Gradient Flow**   As a first step towards deriving the KALE particle descent algorithm, let us first consider a time-discretized version of the KALE gradient flow (Eq. (9) and Eq. (10)), obtained by applying a forward-Euler scheme to Eq. (9) with step size $\gamma$. This time-discretized equation is given by

$$\mathbb{P}_{n+1} = (I - \gamma(1+\lambda)\nabla h_n^\star)_\# \mathbb{P}_n, \ \mathbb{P}_{n=0} = \mathbb{P}_0. \tag{13}$$

The function $h_n^\star$ is a discrete time analogue of Eq. (10), in that it is solution to the following optimization problem:

$$h_n^\star = \arg\max_{h \in \mathcal{H}} \left\{ 1 + \int h \mathrm{d}\mathbb{P}_n - \int e^h \mathrm{d}\mathbb{Q} - \frac{\lambda}{2} \|h\|^2 \right\}. \tag{14}$$

The solution $\mathbb{P}_n$ of Eq. (13) is a sensible approximation of $\mathbb{P}_t$: indeed, it can be shown under suitable smoothness assumptions [54, 4] that the piecewise-constant trajectory $(t \longmapsto \mathbb{P}_n$ if $t \in [n\gamma, (n+1)\gamma))$ obtained from the time-discretized gradient flow of a functional $\mathcal{F}$ will recover the true gradient flow solution $\mathbb{P}_t$ of $\mathcal{F}$ as $\gamma \to 0$.

**Approximation using finitely many samples: the KALE particle descent algorithm**   Despite its discrete-time nature, the sequence $(\mathbb{P}_n)_{n \geq 0}$ may still be intractable to compute: for generic $\mathbb{P}_0$ and $\mathbb{Q}$, Eq. (14) will contain intractable expectations and have an infinite dimensional search space. To address this issue, we propose the *KALE particle descent algorithm*: this algorithm *approximates* the true time-discrete iterates $\mathbb{P}_n$ given $N$ samples $\{X^{(i)}\}_{i=1}^N$ and $\{Y_0^{(i)}\}_{i=1}^N$ of $\mathbb{Q}$ and $\mathbb{P}_0$, by computing the probabilities $\widehat{\mathbb{P}}_n^N$ solving the time-discrete KALE gradient flow arising from the *empirical* source-target pair $\widehat{\mathbb{Q}}^N = \frac{1}{N}\sum_{i=1}^N X^{(i)}$ and $\widehat{\mathbb{P}}_0^N = \frac{1}{N}\sum_{i=1}^N Y_0^{(i)}$. As opposed to $\mathbb{P}_n$, it is possible to *exactly* compute $\widehat{\mathbb{P}}_n^N$: indeed, the recursion equation Eq. (13) implies that $\widehat{\mathbb{P}}_n^N$ remains discrete for all $n$. More precisely, we have $\widehat{\mathbb{P}}_n^N = \frac{1}{N}\sum_{i=1}^N Y_n^{(i)}$, where

$$Y_{n+1}^{(i)} = Y_n^{(i)} - \gamma(1+\lambda)\nabla \widehat{h}_n^\star(Y_n^{(i)}), \tag{15}$$

and $\widehat{h}^\star$ is defined as

$$\widehat{h}_n^\star = \arg\max_{h \in \mathcal{H}} \left\{ \int h \mathrm{d}\widehat{\mathbb{P}}_n^N - \int e^h \mathrm{d}\widehat{\mathbb{Q}}^N - \frac{\lambda}{2} \|h\|_{\mathcal{H}}^2 \right\}. \tag{16}$$

As in the sample-based setting of Eq. (7), $\widehat{\mathbb{P}}_n^N$ and $\widehat{\mathbb{Q}}^N$ are discrete, meaning that Eq. (16) reduces to an $N$- dimensional convex problem, and $\widehat{h}_n^\star$ can be tractably computed. The alternate execution of Eq. (15) and Eq. (16) for a finite number of time steps defines the *KALE Particle Descent Algorithm*, that we lay out in Algorithm 1.

**Consistency of the KALE Particle Descent Algorithm**   Note that the source of error in the KALE particle descent algorithm lies in the use of an approximate witness function $\widehat{h}_n^\star$ instead of the true, but intractable, $h_n^\star$. Indeed, one can show, using the theory of McKean-Vlasov representative processes [39], that the $n$-th iterates of the sequence defined by:

$$\bar{Y}_{n+1}^{(i)} = \bar{Y}_n^{(i)} - \gamma(1+\lambda)\nabla h_n^\star(\bar{Y}_n^{(i)}), \ \bar{Y}_0^{(i)} \sim \mathbb{P}_0, \ 1 \leq i \leq N \tag{17}$$

are distributed according to the $n^{\text{th}}$ iterate $\mathbb{P}_n$ of the true discrete-time KALE gradient flow solution defined in Eq. (13). As such, the discrete probability $\bar{\mathbb{P}}_n^N = \frac{1}{N}\sum_{i=1}^N \delta_{\bar{Y}_n^{(i)}}$ may be considered as an unbiased space-discretization of Eq. (13). In the next proposition, we show that the iterates $\widehat{\mathbb{P}}_n^N$ returned by the KALE particle descent algorithm can approximate the unbiased $\bar{\mathbb{P}}_n^N$ with arbitrarily low error.

**Proposition 4** (Consistency of the KALE particle descent). *Let $\{Y_0^{(i)}\}_{i=1}^N \sim \mathbb{P}_0$. Let $(\bar{\mathbb{P}}_n^N)_{n \geq 0}$ be the sequence of discrete probabilities arising from Eq. (17) with initial conditions $\{Y_0^{(i)}\}_{i=1}^N$, and let $(\widehat{\mathbb{P}}_n^N)_{n \geq 0}$ be the sequence arising from Eq. (15) with the* same *initial conditions $\{Y_0^{(i)}\}_{i=1}^N$. Let $n_{max} \geq 0$. Then, under Assumptions 1 and 2, for all $n \leq n_{\max}$, the following bound holds:*

$$\mathbb{E}W_2(\widehat{\mathbb{P}}_n^N, \bar{\mathbb{P}}_n^N) \leq \frac{A}{B\sqrt{N}}(e^{\gamma B n_{\max}} - 1)$$

*with $A = \sqrt{2KK_{1d}(1 + e^{\frac{8K}{\lambda}})} \times \frac{1}{4\sqrt{KK_{1d}} + K_{2d}}$ $B = \frac{(1+\lambda)(4\sqrt{KK_{1d}} + \sqrt{K_{2d}})}{\lambda}$, and $K, K_{1d}, K_{2d}$ are the constants defined in Assumptions 1 and 2.*

Proposition 4 shows that given a finite time horizon $n_{\max}$, and given sufficiently many samples of $\mathbb{P}_0$ and $\mathbb{Q}$, one can approximate an exact discrete KALE flow between $n = 0$ and $n = n_{\max}$ with arbitrary precision. The proof of Proposition 4 (given in Appendix F) relies on the regularity of the KALE witness function $x \longmapsto \widehat{h}_n^\star(x)$, but also on the regularity of the mapping $\widehat{\mathbb{P}}_n^N \longmapsto \widehat{h}_n^\star$ (using the 2-Wasserstein distance as the metric on $\mathcal{P}_2(\mathbb{R}^d)$).

---

**Algorithm 1** KALE Particle Descent Algorithm

---

**Input:** $\{Y_0^{(i)}\}_{i=1}^N \sim \mathbb{P}_0$, $\{X^{(i)}\}_{i=1}^N \sim \mathbb{Q}$, `max_iter`, $\lambda, k, \gamma$
**Output** $\{Y_{\texttt{max\_iter}}^{(i)}\}_{i=1}^N$
**for** $n = 0$ **to** `max_iter`$-1$ **do**
   `f_star` $\leftarrow$ `dual_solve`$(X^{(1)}, Y_i^{(1)}, \ldots, X^{(N)}, Y_i^{(N)}, k, \lambda)$          `# See Eq.6`
   `h_star` $\leftarrow$ `compute_log_ratio`$(\texttt{f\_star}, X^{(1)}, Y_i^{(1)}, \ldots, X^{(N)}, Y_i^{(N)}, k, \lambda)$   `# Ditto`
   **for** $j = 1$ **to** $N$ **do**
      `v` $\leftarrow (1 + \lambda)$`grad`$(\texttt{h\_star}(Y_i^{(j)}))$
      $Y_{i+1}^{(j)} \leftarrow Y_i^{(j)} - \gamma \times$`v`
   **end for**
**end for**

---

### 4.2 Regularization of KALE particle descent using Noise Injection

In practice, guaranteeing the convergence of the KALE gradient flow (and its corresponding KALE particle descent) by relying on the condition given in Proposition 3 is cumbersome for two reasons: first, this condition is hard to check, and second, it does not tell us what to do when the condition is not met. Noise injection [4, 11] is a practical regularization technique originally introduced for the MMD flow, that trades off some of the "steepest descent" property of gradient flow trajectories with some additional smoothness (in negative Sobolev norm) in order to improve convergence to the target trajectory. We recall that the solution of a (discrete time) noise injected gradient flow with velocity field $(1 + \lambda)\nabla h_n^\star$ and noise schedule $\beta_n$ is defined as the sequence $(\mathbb{P}_n)_{n \geq 0}$ whose iterates verify:

$$\mathbb{P}_{n+1} = ((x, u) \longmapsto x - \gamma(1 + \lambda)\nabla h_n^\star(x + \beta_n u))_\# (\mathbb{P}_n \otimes g), \tag{18}$$

where $g$ is a standard unit Gaussian distribution. As we show in the next proposition, under a suitable noise schedule, noise injection can also be applied to ensure global convergence of the KALE flow.

**Proposition 5** (Global Convergence under noise injection dynamics). *Let $\mathbb{P}_n$ be defined as Eq. (18). Let $(\beta_n)_{n \geq 0}$ be a sequence of noise levels, and define $\mathcal{D}_{\beta_n, \mathbb{P}_n} = \mathbb{E}_{y \sim \mathbb{P}_n, u \sim g} \|\nabla h_n^\star(x + \beta_n u)\|^2$ with $g$ the density of a standard Gaussian distribution. Then, under Assumptions 1 and 2, and for a choice of $\beta_n$ such that:*

$$\frac{8K_{2d}\beta_n^2}{\lambda^2} KALE(\mathbb{P}_n \,||\, \mathbb{Q}) \leq \mathcal{D}_{\beta_n, \mathbb{P}_n}(\mathbb{P}_n),$$

*the following holds: $KALE(\mathbb{P}_{n+1} \,||\, \mathbb{Q}) - KALE(\mathbb{P}_n \,||\, \mathbb{Q}) \leq -\frac{\gamma}{2}(1 - 3\gamma\sqrt{KK_{2d}})D_{\beta_n, \mathbb{P}_n}(\mathbb{P}_n)$. Moreover, if $\sum_{i=1}^\infty \beta_i = +\infty$, then $\lim_{n \to \infty} KALE(\mathbb{P}_n \,||\, \mathbb{Q}) = 0$.*

As in [4], convergence of the regularized KALE flow is guaranteed when the noise schedule satisfies an inequality for all $n$, which is hard to check in practice. Nonetheless, we empirically observe that in all our problems a small, constant noise schedule can help the KALE flow reach a lower KALE value at convergence.

Let us stress that the noise injection scheme given in Proposition 5 is a *population* scheme that includes an intractable convolution. To use noise injection in the KALE particle descent algorithm, we approximate this convolution using a single sample $U_n^{(i)}$ for each particle update. Eq. (15) becomes:

$$Y_{n+1}^{(i)} = Y_n^{(i)} - \gamma(1 + \lambda)\nabla\widehat{h}_n^\star(Y_n^{(i)} + \beta_n U_n^{(i)}), \;\; U_n^{(i)} \sim \mathcal{N}(0, 1). \tag{19}$$

**Implementation**    The particle descent algorithm can be implemented using automatic differentiation software such as the `pytorch` library in python. This allows us to easily compute the gradient of the log-density ratio estimate $\widehat{h}_n^\star$ appearing in the particle update rule Eq. (15).

Computing $\widehat{h}_n^\star$ can be achieved using methods such as gradient descent, coordinate descent or higher order optimization methods such as Newton's method and L-BFGS [35].

## 5    Experiments

In this section, we empirically study the behavior of the KALE particle descent algorithm in three settings reflecting different topological properties for the source-target pair: a pair with a target supported on a hypersurface (zero volume support), a pair with disjoint supports of positive volume, and a pair of distributions with a positive density supported on $\mathbb{R}^d$.

**KALE flow for targets defined on hypersurfaces** Our first example consists in a target *supported* (and uniformly distributed) on a lower-dimensional surface that defines three non-overlapping rings. The initial source is a Gaussian distribution with a mean in the vicinity of the target $\mathbb{Q}$. This setting is a perfect candidate to illustrate the failure modes of both the KL and the MMD when used in particle descent algorithms: on the one hand, the measures $\mathbb{P}_0$ and $\mathbb{Q}$ are mutually singular, and thus the KL gradient flow from $\mathbb{P}_0$ to $\mathbb{Q}$ does not exist. By contrast, the KALE is well-defined in this case, and inherits from the KL an increased sensitivity to support discrepancy. For that reason, we hypothesize that the trajectories of the KALE flow will converge towards a better limit compared to its MMD flow counterpart. We sample $N = 300$ points from the target and the initial source distribution and run an implementation of Algorithm 1 for $n = 50000$ iterations. The complete set of parameters is given in the appendix. Results are plotted in Fig. 1. We indeed notice that the KALE flow trajectory remains

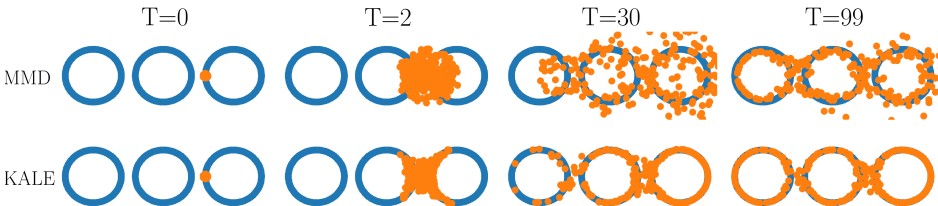

Figure 1: MMD and KALE flow trajectories for "three rings" target

close to the target support and recovers the target almost perfectly. This illustrates the ability of the KALE flow to *relax* the hard support-sharing constraints of the KL flow into soft support closeness constraints. These soft constraints are not present in the MMD flow, where particles of the source can remain scattered around the plane.

**KALE flow between probabilities with disjoint support** In our second example, we consider a source/target pair that are supported on disjoint subsets each with a finite, positive volume (unlike the previous example). The support of the source and the target consist respectively of a heart and a spiral, and the two distributions have a uniform density on their support. Again, because the supports of the source and the target are disjoint, the KL flow cannot be defined, nor simulated for this pair. We run a KALE particle descent algorithm, and compare it as before with an MMD flow, as well as with a "Sinkhorn descent algorithm" [25]. Results are in Fig. 2.

As we can see, the soft support-sharing constraint informing the KALE flow forces the source to quickly recover the spiral shape, much before the Sinkhorn and MMD flow trajectories. However, compared to Sinkhorn, the two KALE-generated spirals have a harder time recovering outliers, disconnected from the main support of the spiral.

**KALE flow for probabilities with densities** We consider the setting where the target admits a positive density on $\mathbb{R}^d$. Hence, unlike in the two previous examples, the KL gradient flow is well-defined, and can be simulated using the Unadjusted Langevin Algorithm (ULA). Echoing the interpolation property of the KALE between the MMD and the KL shown in Proposition 1, we propose to investigate whether this property is preserved in a *gradient flow* setting. We consider a

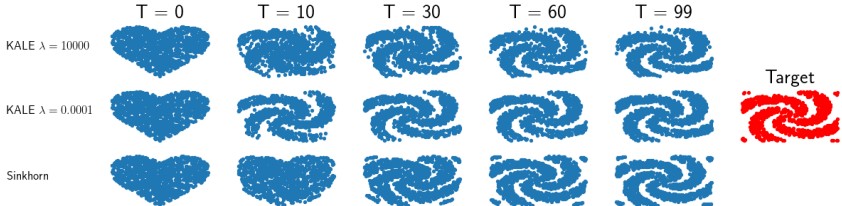

Figure 2: Shape Transfer using the KALE flow

balanced mixture of 4 Gaussians with means located on the 4 corners of the unit square for the target and a source distribution given by a unit Gaussian in the vicinity of the unit square. We then run KL, MMD, and KALE flows with different values of $\lambda$, and compute the Wasserstein distance between reference particles at iteration $n$ from either the MMD or KL flow and particles obtained from the KALE flow at the same iteration $n$. The choice of the Wasserstein distance is natural for Wasserstein Gradient Flows. As shown in Fig. 3a, for "small" values of $\lambda$, particles from a KALE flow remain close to the ULA particles, while for "large" ones they remain close to the MMD particles (Fig. 3b).

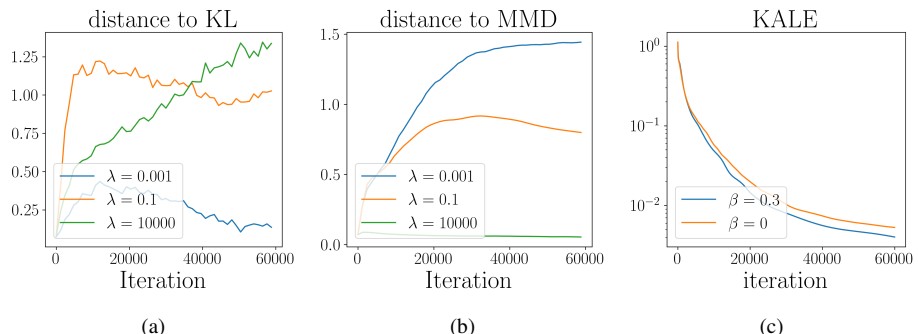

Figure 3: (a): Evolution of the Wasserstein distance between reference particles from the ULA algorithm and the KALE particle descent algorithm with various values of $\lambda$. (b) Left: same as (a), but taking particles from the MMD flow as reference. (c) Evolution of the KALE along the trajectories of a KALE descent algorithm with the same mixture of Gaussians as target, using $\lambda = 0.1$. Orange: without noise injection. Blue: with noise injection using a constant noise schedule.

**Impact of noise injection** On all three examples, using a regularized KALE flow with an appropriately tuned $\beta_n$ schedule always improves the proximity to the global minimum $\mathbb{P}_\infty = \mathbb{Q}$. Its effect is particularly impactful in the mixture of Gaussians example, where a small, constant noise schedule $\beta_n$ allows for faster mixing times for $\mathbb{P}_n$, as opposed to its unregularized counterpart, see Fig. 3c. We provide further details on the impact of noise injection in the appendix.

## 6 Discussion and further work

We have constructed the KALE flow, a gradient flow between probability distributions that relaxes the KL gradient flow for probabilities with disjoint support. Using the *KALE Particle Descent Algorithm*, we have shown on several examples that in cases where a KL gradient flow cannot be defined, trajectories of the KALE flow empirically exhibit better convergence properties when compared to the MMD flow, a flow that the KALE is also able to interpolate. In cases where the KL flow can be defined, we notice empirically that the KALE flow can *approximate* the trajectories of the KL flow, but using only information from *samples* of the target. This latter property is in sharp contrast with KL Gradient Flow discretizations like the Unadjusted Langevin Algorithm: in this regard, we could use the KALE flow as a sample-based approximation of the KL flow, which is to our knowledge a novel concept. Future work would analyze when the KALE flow is a consistent estimator of the KL flow in the large sample limit.

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
