**Appendix for** *KALE flow: A relaxed KL Gradient Flow for Probabilities with Disjoint Support*

The appendix is structured as follows: in Appendix A, we give additional details on the variational formulation of the KL divergence as well as Wasserstein gradient flows. In Appendix B, C, we give the proofs for all statements made about the static properties of the KALE, while in Appendix D to F we give proofs for all statements made about the KALE flow and descent algorithm. Appendix G contains some additional technical lemmas that are used throughout the appendix. Finally, in Appendix H, we provide details on the experiments discussed in the main body, and the impact of noise injection on KALE particle descent trajectories.

## A   Mathematical Background

In this section, we lay out in more depth the theoretical framework behind the tools used in this paper. We first review the variational formulation of the KL, and more generally $f$-divergences. We discuss how this variational formulation can be used beyond the context of statistical estimation of the KL, which is the original context it was considered for [48]. We then provide additional details about Wasserstein gradient flows, and the theoretical tools used to study them.

### A.1   The use of the variational formulation of $f$-divergences

$f$-divergences, first described in [2], form a family of divergences between probability measures parametrized by a convex, lower semi-continuous function $f$. The divergence $D_f$ between two probabilities measures $\mathbb{P}$ and $\mathbb{Q}$ is defined as:

$$D_f(\mathbb{P} \,||\, \mathbb{Q}) = \begin{cases} \int f(\frac{d\mathbb{P}}{d\mathbb{Q}})d\mathbb{Q} & \text{if } \mathbb{P} \ll \mathbb{Q} \\ +\infty & \text{otherwise} \end{cases}$$

Apart from the KL, which we will discuss later, other well known instances of $f$-divergences include the $\chi^2$ divergence, the Hellinger divergence and the Total Variation. Requiring the function $f$ to be convex allows to use the theory of Fenchel duality to frame $D_f$ as the solution of an optimization problem:

**Proposition 6** ([3, Lemma 9.4.4]). *For any $\mathbb{P}$, $\mathbb{Q} \in \mathcal{P}(\mathbb{R}^d)$, we have:*

$$D_f(\mathbb{P} \,||\, \mathbb{Q}) = \sup_{h \in C_b^0(\mathbb{R}^d)} \left\{ \int_{\mathbb{R}^d} h(x)d\mathbb{P} - \int f^\star(h(x))d\mathbb{Q} \right\} \tag{20}$$

Where $f^\star$ is the Fenchel convex conjugate [51] of the convex function $f$, defined as:

$$f^\star(u) = \sup_{x \in \mathbb{R}^d} \langle u, x \rangle - f(x)$$

The KL divergence is a particular instance of $f$-divergence using the pair $(f, f^\star)$:

$$f(x) = \begin{cases} x(\log x - 1) + 1 & \text{if } x > 0 \\ 1 & \text{if } x = 0 \,, \\ +\infty & \text{if } x < 0 \end{cases} \qquad f^\star(u) = e^u - 1$$

**M-estimation procedures for KL**$(\mathbb{P} \,||\, \mathbb{Q})$   The dual formulation in Eq. (20) is an optimization problem with an objective depending on $\mathbb{P}$ and $\mathbb{Q}$ only through *expectations*. By relying on the theory of M-estimation, [48] showed that it was possible to *consistently* approximate the population solution of Eq. (20) using only samples $\{Y^{(i)}\}_{i=1}^N$ and $\{X^{(i)}\}_{i=1}^N$ of $\mathbb{P}$ and $\mathbb{Q}$. In particular, they showed that the solution of the sample-based, regularized problem:

$$\sup_{h \in \mathcal{H}} \left\{ 1 + \int h d\widehat{\mathbb{P}}^N - \int e^h d\widehat{\mathbb{Q}}^N + 1 - \frac{\lambda_N}{2} I(h) \right\} \tag{21}$$

(where $I(h)$ is a convex complexity penalty) will converge in probability to the solution of Eq. (20), provided that $\lambda_N$ decays to 0 as $\frac{1}{\sqrt{N}}$ and that the complexity of the function class $\mathcal{H}$ is small enough. However, their setting is general and does not exploit the specificity of an RKHS $\mathcal{H}$ with a penalty $I(h) = \|h\|_{\mathcal{H}}^2$. Consistency for the latter case (a case which is tightly linked to the definition of the KALE), was proved by [6] using tools from RKHS theory.

**Why KALE differs from simple KL estimation**  The addition of the regularization term $\frac{\lambda_N}{2} I(h)$ (where the KALE objective is retrieved using $I(h) = \|h\|^2$) to Eq. (21) makes the solution of Eq. (20) non-infinite for the mutually singular empirical distributions $\widehat{\mathbb{P}}^N$ and $\widehat{\mathbb{Q}}^N$. However, the KL population objective Eq. (20) is unregularized, reflecting the fact that the KL is infinite for mutually singular population $\mathbb{P}$ and $\mathbb{Q}$. It is the goal of Section 2 is to show that extending the regularization technique introduced in an estimation setting to the KL population objective results in a relaxed solution to the KL problem that is a valid divergence measure between $\mathbb{P}$ and $\mathbb{Q}$. The KALE thus leverages the *biases* of the KL estimates to remain well-defined for mutually singular distributions: in the present context, the primary interest of KALE is not to estimate the KL, but to provide a *KL alternative* for mutually singular distributions. This justifies the definition of the KALE with a positive $\lambda$ given in Definition 1. Note that a sample-based approximation of KALE($\mathbb{P} \parallel \mathbb{Q}$) is now:

$$(1 + \lambda) \max_{h \in \mathcal{H}} \left\{ 1 + \int h \, d\widehat{\mathbb{P}}^N - \int e^h \, d\widehat{\mathbb{Q}}^N - \frac{\lambda}{2} \|h\|^2 \right\} \tag{22}$$

We emphasize that unlike in Eq. (21), $\lambda$ is now kept fixed.

## A.2 Wasserstein Gradient Flows

**The Wasserstein Geometry**  The theory of Wasserstein-2 gradient flows considers the set of probability measures on $\mathcal{P}_2(\mathcal{X})$ (where $\mathcal{X}$ is a separable Hilbert Space set to $\mathbb{R}^d$ in our case) with finite 2$^{\text{nd}}$ order moments, endowed with the Wasserstein-2 metric, defined, given $\mathbb{P}_0, \mathbb{P}_1 \in \mathcal{P}_2(\mathbb{R}^d)$, as:

$$W_2(\mathbb{P}_0, \mathbb{P}_1) = \left( \inf_{\gamma \in \Gamma(\mathbb{P}_0, \mathbb{P}_1)} \int \|x - y\|^2 \, d\gamma(x, y) \right)^{\frac{1}{2}} \tag{23}$$

$\Gamma(\mathbb{P}_0, \mathbb{P}_1)$ denotes the sets of *admissible transport plans* between $\mathbb{P}_0$ and $\mathbb{P}_1$:

$$\Gamma(\mathbb{P}_0, \mathbb{P}_1) = \left\{ \gamma \in \mathcal{P}(\mathbb{R}^d \times \mathbb{R}^d); \quad (\pi^1)_\# \gamma = \mathbb{P}_0, \ (\pi^2)_\# \gamma = \mathbb{P}_1 \right\}$$

where $\pi^1 : (x, y) \longmapsto x$ and $\pi^2(x, y) \longmapsto y$ are the canonical projections on $\mathbb{R}^d \times \mathbb{R}^d$. In the proofs, we will often consider *constant speed geodesics* between two probabilities $\mathbb{P}_0$ and $\mathbb{P}_1$, defined as paths $(\mathbb{P}_t)_{0 \leq t \leq 1}$ of the form:

$$\mathbb{P}_t = \left( (1 - t)\pi^1 + t\pi^2 \right)_\# \gamma$$

where $\gamma \in \Gamma_o(\mathbb{P}_0, \mathbb{P}_1)$ is an *optimal coupling*, in the sense that it minimizes the objective defining the $W_2(\mathbb{P}_0, \mathbb{P}_1)$ distance in Eq. (23). Convexity along geodesics, or *geodesic convexity* is a property of functionals in $(\mathcal{P}_2(\mathbb{R}^d), W_2)$:

**Definition 2** (Geodesic convexity, [3, Definition 9.1.1]). *We say that a functional $\mathcal{F}$ is $-M$-geodesically semiconvex for some $M > 0$ if for any $\mathbb{P}_0, \mathbb{P}_1$ and constant speed geodesic $\mathbb{P}_t$, $t \in [0, 1]$ between $\mathbb{P}_0$ and $\mathbb{P}_1$, the following holds:*

$$\mathcal{F}(\mathbb{P}_t) \leq (1 - t)\mathcal{F}(\mathbb{P}_0) + t\mathcal{F}(\mathbb{P}_1) + Mt(1 - t) W_2(\mathbb{P}_0, \mathbb{P}_1)^2.$$

**Wasserstein Gradient Flows**  The set $(\mathcal{P}_2(\mathbb{R}^d), W_2)$ is a metric space and not a Hilbert space. Because of that, the notion of gradient (flow) of a functional $\mathcal{F}$ cannot easily be defined through duality with the differential of $\mathcal{F}$, and porting the notion of "gradient flow" to the space $(\mathcal{P}_2(\mathbb{R}^d), W_2)$ thus requires characterizing gradient flows trajectories in a Hilbertian-free way. Examples of such characterizations include curves of maximal slope [3, Section 11.1.1], or identification with limit curves of *minimizing moment schemes*. We refer to [55] for an introduction of gradient flows Wasserstein spaces. The formal definition of Wasserstein-2 gradient flows as given in [3] is as follows:

**Definition 3** (Gradient Flows [3, Definition 11.1.1]). *We say that an absolutely continuous map $\left( t \longmapsto \mathbb{P}_t \in \mathcal{P}_2(\mathbb{R}^d) \right)$ is a solution of the* Wasserstein-2 gradient flow *equation:*

$$\partial_t \mathbb{P}_t + div\left( \mathbb{P}_t v_t \right) = 0, \tag{24}$$

*if $(I \times (-v_t)) \in \boldsymbol{\partial}\mathcal{F}(\mathbb{P}_t)$, where $\boldsymbol{\partial}\mathcal{F}(\mathbb{P}_t)$ is the extended Fréchet subdifferential of $\mathcal{F}$ evaluated at $\mathbb{P}_t$.*

For common functionals such as the sum of (sufficiently smooth) potential, interaction and internal energy terms,

$$\mathcal{F}(\mathbb{P}) = \int V(x)d\mathbb{P}(x) + \int W(x-y)d\mathbb{P}(x)d\mathbb{P}(y) + \int f(p(x))d\mathbb{P}(x)$$

(where $\mathbb{P}$ is assumed to be regular, of the form $d\mathbb{P}(x) = p(x)\mathrm{d}x$), [3] have identified solutions of the very general Eq. (24) with solutions of the more familiar

$$\partial\mathbb{P}_t - \mathrm{div}\left(\mathbb{P}_t\nabla\frac{\delta\mathcal{F}}{\delta\mathbb{P}}(\mathbb{P}_t)\right) = 0, \tag{25}$$

where $\frac{\delta\mathcal{F}}{\delta\mathbb{P}}$ is the first variation of $\mathcal{F}$, defined (when it exists) as the function $v$ verifying:

$$\lim_{\epsilon\to 0}\frac{\mathcal{F}(\mathbb{P}+\epsilon d\chi) - \mathcal{F}(\mathbb{P})}{\epsilon} = \int v(x)\mathrm{d}\chi, \quad \chi = \mathbb{P} - \mathbb{Q}$$

For any $\mathbb{Q} \in \mathcal{P}_2(\mathbb{R}^d)$. Note that this case includes the MMD (given regularity assumptions on the kernel), as discussed in [4], but does not include the KALE, which is not a functional studied in [3], and to our knowledge, a novel object of study in the Wasserstein gradient flow literature. In Appendix D, we show that the identification between Eq. (24) and Eq. (25) still holds for the case of KALE, by identifying elements of its (strong) *extended Fréchet subdifferential*. For completeness, we recall the definition of a strong extended Fréchet subdifferential:

**Definition 4** ((Strong) Extended Fréchet subdifferential, [3, Definition 10.3.1]). *Let $\mathcal{F} : \mathcal{P}_2(\mathbb{R}^d) \longmapsto (-\infty, +\infty]$ be a proper, geodesically convex functional that is lower semicontinuous w.r.t $W_2$. We say that $\gamma \in \mathcal{P}_2(\mathbb{R}^d \times \mathbb{R}^d)$ belongs to the strong extended Fréchet subdifferential $\partial F(\mathbb{P}_0)$ if $\left(\pi^1\right)_{\#}\gamma = \mathbb{P}_0$, and for every $\mathbb{P}_1 \in \mathcal{P}_2(\mathbb{R}^d)$ and $\boldsymbol{\mu} \in \Gamma(\gamma, \mathbb{P}_1)$:*

$$\mathcal{F}(\mathbb{P}_1) - \mathcal{F}(\mathbb{P}_0) \geq \int_{X^3}\langle x_2, x_3 - x_1\rangle d\boldsymbol{\mu} + o(W_{2,\boldsymbol{\mu}}(\mathbb{P}_0, \mathbb{P}_1))$$

*where $W_{2,\boldsymbol{\mu}}^2(\mathbb{P}_0, \mathbb{P}_1) = \int \|x_1 - x_3\|^2 d\boldsymbol{\mu}(x_1, x_2, x_3)$.*

## B  Proof of Theorem 1

Throughout this proof, we will consider the function $\mathcal{K} : \mathcal{H} \times \mathcal{P}(\mathbb{R}^d) \to \mathbb{R}$ given by:

$$\mathcal{K}(h, \mathbb{P}) = \max\left\{1 + \int hd\mathbb{P} - \int e^h d\mathbb{Q} - \frac{\lambda}{2}\|h\|^2\right\} \tag{26}$$

$\mathcal{K}$ has the same expression as the one of Lemma 1, with a supercharged signature to include the dependency in $\mathbb{P}$, which we will use in this proof. Note that the use of $\max$ (and not $\sup$) in the definition of $\mathcal{K}$ is made possible since (minus) the objective of $\mathcal{K}$ is continuous w.r.t $\mathcal{H}$'s strong topology, and convex: thus it is lower-semicontinuous [15, Corollary 3.9]. Following this, one can use the extreme-value theorem to obtain the existence of a maximizer on any RKHS ball (which are weakly compact), and apply a coercivity argument to conclude on the existence of a global maximizer $h^\star$. for any $\mathbb{P}$, the existence of an

**Proof of Lemma 1**  The proof follows directly from [6, Lemma 8]. By Assumption 1, all integrability requirements are satisfied (the two *Bochner* integrals in the next equation are well-defined because of Assumption 1). Following this, the gradient of $\mathcal{K}$ is given by:

$$\nabla_h\mathcal{K}(h, \mathbb{P}) = \int k(x, \cdot)d\mathbb{P} - \int k(x, \cdot)e^h d\mathbb{Q} - \lambda h.$$

And its evaluation at 0 given in Lemma 1 follows. $\qquad\square$

**Proof that the KALE is weakly continuous**  Let $(\mathbb{P}_n)_{n\in\mathbb{N}}$ such that $\mathbb{P}_n$ weakly converges to $\mathbb{P}$. Let $h^\star = \arg\max_h \mathcal{K}(h, \mathbb{P})$ and $h_n^\star = \arg\max_h \mathcal{K}(h, \mathbb{P}_n)$.

$$\limsup_{n\to\infty}\mathcal{K}(h_n^\star, \mathbb{P}_n) = \mathcal{K}(h^\star, \mathbb{P}) \quad \text{and} \quad \liminf_{n\to\infty}\mathcal{K}(h_n^\star, \mathbb{P}_n) = \mathcal{K}(h^\star, \mathbb{P}).$$

The result on KALE follows since when $\lambda$ is kept fixed, $\mathcal{K}$ and KALE differ only by a multiplicative factor. We focus on proving the first (lim sup) equality, the arguments for lim inf being identical.

First, by optimality of $h_n^\star$ w.r.t $\mathbb{P}_n$, we have: $\mathcal{K}(h_n^\star, \mathbb{P}_n) \geq \mathcal{K}(h^\star, \mathbb{P}_n)$, implying

$$\limsup_{n \to \infty} \mathcal{K}(h_n^\star, \mathbb{P}_n) \geq \limsup_{n \to \infty} \mathcal{K}(h^\star, \mathbb{P}_n).$$

Since $\mathbb{P}_n \rightharpoonup \mathbb{P}$, the r.h.s verifies $\limsup_{n \to \infty} \mathcal{K}(h^\star, \mathbb{P}_n) = \lim_{n \to \infty} \mathcal{K}(h^\star, \mathbb{P}_n) = \mathcal{K}(h^\star, \mathbb{P})$, from which we conclude $\limsup_{n \to \infty} \text{KALE}(\mathbb{P}_n, \mathbb{Q}) \geq \text{KALE}(\mathbb{P} \mid\mid \mathbb{Q})$. To prove the converse, assume that $\limsup_{n \to \infty} \text{KALE}(\mathbb{P}_n, \mathbb{Q}) > \text{KALE}(\mathbb{P} \mid\mid \mathbb{Q})$. Then there exists $\epsilon > 0$ and a subsequence $n_k \to +\infty$ with $k \to +\infty$ such that $\mathcal{K}(h_{n_k}^\star, \mathbb{P}_{n_k}) \geq \mathcal{K}(h^\star, \mathbb{P}) + \frac{\epsilon}{2}$. Let us now compare $\mathcal{K}(h_{n_k}^\star, \mathbb{P})$ with $\mathcal{K}(h^\star, \mathbb{P})$ :

$$\mathcal{K}(h_{n_k}^\star, \mathbb{P}) = \mathcal{K}(h_{n_k}^\star, \mathbb{P}_{n_k}) + \int h_{n_k}^\star \mathrm{d}(\mathbb{P} - \mathbb{P}_{n_k}) \geq \mathcal{K}(h^\star, \mathbb{P}) + \frac{\epsilon}{2} - \frac{4\sqrt{K}\text{MMD}(\mathbb{P} \mid\mid \mathbb{P}_{n_k})}{\lambda}$$

where for the last step, we used the Cauchy-Schwarz inequality and Lemma 5. Since the MMD is weakly continuous for bounded kernels with Lipschitz embeddings [59, Theorem 3.2], we have $\lim_{k \to \infty} \text{MMD}^2(\mathbb{P}_{n_k} \mid\mid \mathbb{P}) = 0$: there exists a $k_0$ such that, for $k > k_0$, $\mathcal{K}(h_{n_k}^\star, \mathbb{P}) > \mathcal{K}(h^\star, \mathbb{P}) + \frac{\epsilon}{4}$, which contradicts the optimality condition defining $h^\star$. Hence, we must have

$$\limsup_{n \to \infty} \text{KALE}(\mathbb{P}_n \mid\mid \mathbb{Q}) = \text{KALE}(\mathbb{P} \mid\mid \mathbb{Q}).$$

The two steps of this proof can be repeated for any convergent subsequence of $\mathcal{K}(h_n^\star, \mathbb{P}_n)$, and as a consequence, we also have: $\liminf_{n \to \infty} \text{KALE}(\mathbb{P}_n \mid\mid \mathbb{Q}) = \text{KALE}(\mathbb{P} \mid\mid \mathbb{Q})$, which proves the weak continuity of KALE. $\qquad\square$

**Proof that KALE is a probability divergence that metrizes the weak convergence of probability distributions** We first prove positivity and definiteness of KALE, making it a probability divergence. Positivity of KALE comes from the fact that $\mathcal{K}(h^\star, \mathbb{P}) \geq \mathcal{K}(0, \mathbb{P}) = 0$. To prove definiteness of KALE, assume $\text{KALE}(\mathbb{P} \mid\mid \mathbb{Q}) = 0$. Recall that $\text{KALE}(\mathbb{P} \mid\mid \mathbb{Q}) = 0 \iff h^\star = 0$, since $\mathcal{K}(0, \mathbb{P}) = 0$ and the objective is strongly convex. The optimality criterion for $0_{\mathcal{H}}$ can be characterized by differentiating $\mathcal{K}(h, \mathbb{P})$. Using Lemma 1, and the optimality of 0, we have:

$$0 = \nabla_h \mathcal{K}(0, \mathbb{P}) \triangleq \int k(x, \cdot)\mathrm{d}\mathbb{P} - \int k(x, \cdot)\mathrm{d}\mathbb{Q} = f_{\mathbb{P},\mathbb{Q}},$$

where $f_{\mathbb{P},\mathbb{Q}}$ denotes the MMD *witness function* between $\mathbb{P}$ and $\mathbb{Q}$, i.e. $\text{MMD}(\mathbb{P} \mid\mid \mathbb{Q})^2 = \|f_{\mathbb{P},\mathbb{Q}}\|^2$. When $k$ is universal, $f_{\mathbb{P},\mathbb{Q}} = 0$ is only possible when $\mathbb{P} = \mathbb{Q}$, which proves the first implication of the equivalence. The reverse implication is proven by noticing that

$$\mathbb{P} = \mathbb{Q} \implies \nabla_h \mathcal{K}(0, \mathbb{P}) = 0.$$

**Metrizing weak convergence** From the weak continuity of KALE associated with the definiteness of KALE proven above, we have $\mathbb{P}_n \rightharpoonup \mathbb{Q} \implies \text{KALE}(\mathbb{P}_n \mid\mid \mathbb{Q}) \to 0$. For the converse, assume that $\text{MMD}(\mathbb{P}_n \mid\mid \mathbb{Q})$ doesn't converge to 0. Therefore, there exists a subsequence $n_k$ with $n_k \to +\infty$ when $k \to +\infty$ and such that $\text{MMD}(\mathbb{P}_{n_k} \mid\mid \mathbb{Q}) > c > 0$ for some $c > 0$. Fix $\epsilon > 0$. We have that:

$$\frac{1}{(1+\lambda)}\text{KALE}(\mathbb{P}_{n_k} \mid\mid \mathbb{Q}) \geq \mathcal{K}(\epsilon \times f_{\mathbb{P}_{n_n},\mathbb{Q}}) = \left\langle \nabla_h \mathcal{K}(0, \mathbb{P}_{n_k}), \epsilon f_{\mathbb{P}_{n_k},\mathbb{Q}} \right\rangle + \mathcal{O}(\epsilon^2 \|f_{\mathbb{P}_{n_k},\mathbb{Q}}\|)$$

$$= \epsilon \|f_{\mathbb{P}_{n_k},\mathbb{Q}}\| + \mathcal{O}(\epsilon^2 \|f_{\mathbb{P}_{n_k},\mathbb{Q}}\|^2).$$

Now, recall that $\|f_{\mathbb{P}_{n_k},\mathbb{Q}}\| = \text{MMD}(\mathbb{P}_{n_k} \mid\mid \mathbb{Q}) \geq c > 0$, implying that for sufficiently low $\epsilon$, we will have: $\text{KALE}(\mathbb{P}_{n_k} \mid\mid \mathbb{Q}) > \frac{(1+\lambda)c\epsilon}{2}$, $\forall k \geq n_k$. Thus, $\text{KALE}(\mathbb{P}_n \mid\mid \mathbb{Q})$ does not tend to 0. Hence, by contradiction $\text{MMD}(\mathbb{P}_n \mid\mid \mathbb{Q})$ converges to 0 which implies that $\mathbb{P}_n$ converges weakly to $\mathbb{Q}$ since the MMD metrizes weak convergence. This concludes the proof of Theorem 1. $\qquad\square$

# C Proof of Proposition 1

## C.1 Proof of (i)

To prove that KALE converges to the MMD as $\lambda$ increases, we will show the following inequalities:

$$\frac{1}{2}\text{MMD}^2(\mathbb{P} \mid\mid \mathbb{Q}) - \mathcal{O}\left(\frac{1}{\lambda}\right) \leq \text{KALE}(\mathbb{P} \mid\mid \mathbb{Q}) \leq \frac{1}{2}\text{MMD}^2(\mathbb{P} \mid\mid \mathbb{Q}) + \mathcal{O}\left(\frac{1}{\lambda}\right)$$

To prove the right inequality, we recall that $\mathcal{K}(h, \mathbb{P}) \leq \int h\mathrm{d}\mathbb{P} - \int h\mathrm{d}\mathbb{Q} - \frac{\lambda}{2}\|h\|^2$, which holds by convexity of the exponential. The right-hand side is maximized for $h^\star = \frac{f_{\mathbb{P},\mathbb{Q}}}{\lambda}$ and equals $\frac{\text{MMD}^2(\mathbb{P}\mid\mid\mathbb{Q})}{2\lambda}$. Consequently, we have: $\text{KALE}(\mathbb{P} \mid\mid \mathbb{Q}) \leq \frac{1+\lambda}{2\lambda}\text{MMD}^2(\mathbb{P} \mid\mid \mathbb{Q})$.

To prove the left inequality, we use Lemma 5 which allows to control the discrepancy between the KALE and the MMD. Indeed, we have: $h(x) = \langle h, k(x, \cdot) \rangle \leq \sqrt{K}\|h\| = \frac{4K}{\lambda}$. The following Taylor-Lagrange inequality holds, uniformly for all $x$:

$$e^{h(x)} \leq 1 + h(x) + \frac{e^{\frac{4K}{\lambda}}16K^2}{2\lambda^2},$$

which gives a lower bound of $\mathcal{K}(h^\star, \mathbb{P})$:

$$(1 + \lambda)\mathcal{K}(h, \mathbb{P}) \geq (1 + \lambda)\left(\int h\mathrm{d}\mathbb{P} - \int h\mathrm{d}\mathbb{Q} - \frac{\lambda}{2}\|h\|^2 - \frac{8K^2 e^{\frac{4K}{\lambda}}}{\lambda^2}\right).$$

Remark that the r.h.s is maximized for $h_1 = f_{\mathbb{P},\mathbb{Q}}/\lambda$. Because $h^\star$ maximizes the l.h.s, we have:

$$(1 + \lambda)\mathcal{K}(h^\star, \mathbb{P}) \geq (1 + \lambda)\mathcal{K}(h_1, \mathbb{P}) \geq \frac{1 + \lambda}{2\lambda}\text{MMD}^2(\mathbb{P} \mid\mid \mathbb{Q}) - \frac{8K^2 e^{\frac{4K}{\lambda}}(1 + \lambda)}{\lambda^2}.$$

The two initial inequalities are verified, and taking them to the limit $\lambda \to \infty$ concludes the proof. $\square$

## C.2 Proof of (ii)

(ii) was proved in [6] as part of (Theorem 7). For completeness, we recall the elements of the proof. Let us highlight the dependency of $h^\star = \arg\max_h \mathcal{K}(h, \mathbb{P})$ in $\lambda$ (see Eq. (26)) by noting it $h^\star_\lambda(=h^\star)$, for $\lambda \geq 0$. Because we assume that $\log\frac{\mathrm{d}\mathbb{P}}{\mathrm{d}\mathbb{Q}} \in \mathcal{H}$, we have:

$$h^\star_0 = \log\frac{\mathrm{d}\mathbb{P}}{\mathrm{d}\mathbb{Q}}, \quad \text{KL}(\mathbb{P} \mid\mid \mathbb{Q}) = 1 + \int h^\star_0 \mathrm{d}\mathbb{P} - \int e^{h^\star_0}\mathrm{d}\mathbb{Q}.$$

Thus, we have

$$\left|\frac{\text{KALE}(\mathbb{P} \mid\mid \mathbb{Q})}{(1 + \lambda)} - \text{KL}(\mathbb{P} \mid\mid \mathbb{Q})\right| = \left|1 + \int h^\star_\lambda \mathrm{d}\mathbb{P} - \int e^{h^\star_\lambda}\mathrm{d}\mathbb{Q} - \frac{\lambda}{2}\|h^\star_\lambda\|^2_{\mathcal{H}} - \text{KL}(\mathbb{P} \mid\mid \mathbb{Q})\right|$$

$$= \left|\int (h^\star_\lambda - h^\star_0)\mathrm{d}\mathbb{P} - \int e^{h_0}(1 - e^{(h^\star_\lambda - h^\star_0)})\mathrm{d}\mathbb{Q} + \frac{\lambda}{2}\|h^\star_\lambda\|^2\right|$$

$$\leq \left|\int (h^\star_\lambda - h^\star_0)\mathrm{d}\mathbb{P}\right| + \left|\int e^{h_0}(1 - e^{(h^\star_\lambda - h^\star_0)})\mathrm{d}\mathbb{Q}\right| + \left|\frac{\lambda}{2}\|h^\star_\lambda\|^2\right|$$

To bound the last term, we note that

$$\|h^\star_\lambda\| \leq \|h^\star_0\|. \tag{27}$$

Otherwise, by optimality of $h^\star_0$, we have:

$$\int h^\star_\lambda \mathrm{d}\mathbb{P} - \int e^{h^\star_\lambda}\mathrm{d}\mathbb{Q} \leq \int h^\star_0 \mathrm{d}\mathbb{P} - \int e^{h^\star_0}\mathrm{d}\mathbb{Q}$$

$$\implies \int h^\star_\lambda \mathrm{d}\mathbb{P} - \int e^{h^\star_\lambda}\mathrm{d}\mathbb{Q} - \frac{\lambda}{2}\|h^\star\|^2_{\mathcal{H}} \leq \int h^\star_0 \mathrm{d}\mathbb{P} - \int e^{h^\star_0}\mathrm{d}\mathbb{Q} - \frac{\lambda}{2}\|h^\star_0\|^2_{\mathcal{H}},$$

contradicting the optimality of $h^\star_\lambda$. As a consequence, we have that $\lim_{\lambda\to 0}\frac{\lambda}{2}\|h^\star_\lambda\|^2(\leq \frac{\lambda}{2}\|h^\star_0\|^2) = 0$. To bound the first two terms, we use [6] (Lemma 11), ensuring that:

$$\lim_{\lambda\to 0}\|h^\star_\lambda - h^\star_0\| = 0. \tag{28}$$

As a consequence:

- For all $x \in \mathbb{R}^d$, $\lim_{\lambda \to 0} h_\lambda^\star(x) - h_0^\star(x) = 0$.
- $h_\lambda^\star$ is a bounded function.

We conclude that the first two terms tend to 0 as $\lambda \to 0$ by the dominated convergence theorem. We thus have: $\lim_{\lambda \to 0} \left| \text{KALE}(\mathbb{P} \mid\mid \mathbb{Q}) - \text{KL}(\mathbb{P} \mid\mid \mathbb{Q}) \right| = 0$. $\qquad \square$

## D  Proof of Proposition 2

As explained in the introduction, the Wasserstein gradient flow of the KALE does not have a known expression, other than the abstract one given by Definition 3, applied to the KALE. Relying on the formalism introduced in [3], we first show that KALE's gradient flow admits the "traditional" form:

$$\partial_t \mathbb{P}_t - \text{div}\left(\mathbb{P}_t \nabla \frac{\delta \text{KALE}}{\delta \mathbb{P}}\right) = 0$$

We start by giving an expression of the *first variation* of the KALE. This proof is the first in the appendix that involves an implicit function theorem argument, which we justify at length. For brevity, the same justifications will be skipped in other proofs relying on small variations around the same implicit function theorem argument.

**Lemma 2** (Differentiability of KALE).
*Let $\mathbb{Q} \in \mathcal{P}_2(\mathbb{R}^d)$, and $\lambda > 0$. Then, the function $\mathbb{P} \in \mathcal{P}_2(\mathbb{R}^d) \longmapsto KALE(\mathbb{P} \mid\mid \mathbb{Q})$ is Gâteaux differentiable w.r.t. $\mathbb{P}$ and admits the following first variation:*

$$\frac{\delta KALE(\mathbb{P} \mid\mid \mathbb{Q})}{\delta \mathbb{P}} = (1 + \lambda) h^\star, \quad h^\star = \arg\max_{h \in \mathcal{H}} \mathcal{K}(h, \mathbb{P}).$$

*Proof.* Informally, computing the first variation of KALE w.r.t $\mathbb{P}$ can be done using a chain rule argument:

$$\frac{\delta \text{KALE}}{\delta \mathbb{P}} = \frac{\delta \text{KALE}(h^\star(\mathbb{P}), \mathbb{P})}{\delta \mathbb{P}} = \frac{\partial \text{KALE}}{\partial \mathbb{P}} + \frac{\partial \text{KALE}}{\partial h}\Bigg|_{h^\star} \frac{\partial h^\star}{\partial \mathbb{P}} = \frac{\partial \text{KALE}}{\partial \mathbb{P}}$$

where the second term is 0 given that $h^\star$ is defined as $\max_{h \in \mathcal{H}} \mathcal{K}(h, \mathbb{P})$. To make this discussion rigorous, we need to make sure that "$\frac{\partial h^\star}{\partial \mathbb{P}}$" (formally, the Gâteaux derivative of the map $\mathbb{P} \longmapsto h^\star(\mathbb{P})$) exists.

We recall that given two topologically convex vector spaces $X$ and $Y$, and a function $f : X \to Y$, the Gâteaux derivative of $f$ at $x$ in the direction $\chi \in X$ is defined as:

$$Df(x; \chi) = \lim_{t \to 0} \frac{f(x + t\chi) - f(x)}{t}.$$

A complete argument for the differentiability of both $\mathcal{K}$ and $\mathbb{P} \longmapsto h^\star(\mathbb{P})$ would require augmenting the domains of functionals of interest from $\mathcal{P}_2(\mathbb{R}^d)$ (which is not a vector space) by the vector space of signed radon measures $\mathcal{M}(\mathbb{R}^d)$. We circumvent this additional step by simply considering "admissible" directions $\chi$, such that $\int d\chi = 0$. Noting $h_t^\star = \arg\max_h \mathcal{K}(h, \mathbb{P} + t\chi)$, we know given Lemma 1 that $h_t^\star$ verifies:

$$\mathcal{F}_\chi(h_t^\star, t) \triangleq \nabla_h \mathcal{K}(h_t^\star, \mathbb{P}) = \int k(x, \cdot) d(\mathbb{P}(x) + t\chi(x)) - \int k(x, \cdot) \exp(h_t^\star(x)) d\mathbb{Q}(x) - \lambda h_t^\star = 0.$$

Thus, $h_t^\star$ is defined *implicitly* through $\mathcal{K}$'s optimality at $h_t^\star$. To study the differentiability of the mapping $t \longmapsto h_t^\star$, it is natural to rely on an implicit function theorem argument on $\mathcal{F} : (\mathcal{H} \times \mathbb{R}) \to \mathcal{H}$. Similarly to implicit function theorems on euclidean spaces, we will need to invert $D_h \mathcal{F}_\chi(h, t)$, the (Fréchet) differential of $\mathcal{F}$ w.r.t $h$. This differential is given by:

$$D_h \mathcal{F}_\chi(h, t) = \underbrace{-\int k(x, \cdot) \otimes k(x, \cdot) e^{h(x)} d\mathbb{Q}(x) - \lambda I}_{\triangleq \boldsymbol{L}(h)},$$

which is an invertible operator on $\mathcal{H}$, given that $\boldsymbol{L}(h)$ is self-adjoint and positive for all $h$. We can now apply an implicit function theorem on Banach spaces [32] (Theorem 5.9): For all $\chi$, there exists a neighborhood of $0$, $\mathcal{V}(0)$, such that the mapping $t \in \mathcal{V}(0) \longmapsto h_t^\star$ is differentiable. The derivative of $h_t^\star$ at $0$ is then the Gâteaux derivative of $h^\star(\mathbb{P})$ in the direction $\chi$:

$$D_\mathbb{P} h^\star(\mathbb{P}; \chi) = \int (\boldsymbol{L}(h^\star) + \lambda I)^{-1} k(x, \cdot) d\chi.$$

To conclude on KALE's first variation, we can rigorously write, using the chain rule of Gâteaux derivatives,

$$D_\mathbb{P}\text{KALE}(\mathbb{P} \,||\, \mathbb{Q}; \chi) = (1 + \lambda) \left\{ \int h^\star(\mathbb{P})(x) d\chi(x) + \underbrace{\langle \nabla_h \mathcal{K}(h^\star(\mathbb{P}), \mathbb{P}), Dh^\star(\mathbb{P}; \chi) \rangle_\mathcal{H}}_{=0} \right\}$$

$$= (1 + \lambda) \int h^\star(\mathbb{P}) d\chi$$

which concludes the proof. $\qquad\square$

We now show that the KALE admits strong Fréchet subgradients, and that they are equal to the gradient of KALE's first variation.

**Lemma 3.** *A coupling $\gamma$ of the form $(I \times v)_\#\mathbb{P}_0$ belongs to the extended (strong) Fréchet subdifferential of KALE at $\mathbb{P} = \mathbb{P}_0$ if and only if $v = \nabla \frac{\delta KALE}{\delta \mathbb{P}} = (1 + \lambda) \nabla h_0^\star$ $\mathbb{P}_0$-a.e, where $(1 + \lambda) h_0^\star = (1 + \lambda) \arg\max_h \mathcal{K}(h, \mathbb{P}_0)$ is the first variation of KALE at $\mathbb{P} = \mathbb{P}_0$.*

*Proof.* Using an analogue of [3, Equation 10.3.13] for the extended *strong* Fréchet subdifferential, we have that:

$$\gamma = (I \times v)_\#\mathbb{P}_0 \in \partial\text{KALE}(\mathbb{P}_0 \,||\, \mathbb{Q}) \iff$$

$$\text{KALE}(\mathbb{P}_1 \,||\, \mathbb{Q}) - \text{KALE}(\mathbb{P}_0 \,||\, \mathbb{Q}) \geq \int (y - x)^\top v(x) d\tilde{\gamma}(x, y) + o(C_2(\tilde{\gamma}))$$

for any $\mathbb{P}_1 \in \mathcal{P}_2(\mathbb{R}^d)$, $\tilde{\gamma} \in \Gamma(\mathbb{P}_0, \mathbb{P}_1)$. Note that without loss of generality, we switched the coupling $\boldsymbol{\mu} \in \Gamma((I \times v)_\#\mathbb{P}_0, \mathbb{P}_1)$ present in Definition 4 with a coupling $\tilde{\gamma} \in \Gamma(\mathbb{P}_0, \mathbb{P}_1)$, a switch that is made possible because of the specific form of $\gamma$ considered above, which is the one needed in Definition 3. Our goal is to show that $(I \times v)_\#\mathbb{P}_0 \in \partial\text{KALE}(\mathbb{P}_0 \,||\, \mathbb{Q}) \iff v = (1 + \lambda)\nabla h_0^\star$. We first show the reverse implication, e.g. $(I \times (1 + \lambda)\nabla h_0^\star)_\#\mathbb{P}_0 \in \partial\text{KALE}(\mathbb{P}_0 \,||\, \mathbb{Q})$. To do so, we consider the following interpolation scheme between $\mathbb{P}_0$ and $\mathbb{P}_1$:

$$\mathbb{P}_t = \left(t\pi^2 + (1-t)\pi^1\right)_\# \tilde{\gamma}.$$

And note for each $\mathbb{P}_t$, $h_t^\star = \arg\max_h \mathcal{K}(h, \mathbb{P}_t)$. Noting $g(t) = \text{KALE}(\mathbb{P}_t \,||\, \mathbb{Q})$, we have:

$$g'(t) = (1 + \lambda) \int (y - x)^\top \nabla h_t^\star(ty + (1-t)x) d\tilde{\gamma}(x, y), \quad g''(t) = (1 + \lambda)((I) + (II))$$

where

$$(I) = \int (y - x)^\top \left(\mathbf{H}h_t^\star(ty + (1-t)x)(y - x)\right) d\tilde{\gamma}(x, y)$$

$$(II) = \int (y - x)^\top \left(\nabla \frac{dh_t^\star}{dt}(ty + (1-t)x)\right) d\tilde{\gamma}(x, y)$$

(and we exchanged the $t$-derivative and $\nabla$ in (II)). From Assumption 2 we have that $\|\mathbf{H}h\| \leq \|h\| \sqrt{K_{2d}} \leq \frac{4\sqrt{KK_{2d}}}{\lambda}$, implying $(I) \leq \frac{4\sqrt{KK_{2d}}}{\lambda} C_2^2(\tilde{\gamma})$. Using an implicit function theorem argument, we have:

$$\frac{dh_t^\star}{dt} = -(\boldsymbol{L}(h_t^\star) + \lambda I)^{-1}(y - x)^\top \nabla_1 k_{ty+(1-t)x},$$

implying

$$(II) = \int \left\langle \sum_{i=1}^d (y_i - x_i)\partial_i k_{ty+(1-t)x}, \sum_{i=1}^d (y_i - x_i)(\boldsymbol{L}(h_t^\star) + \lambda I)^{-1}\partial_i k_{ty+(1-t)x} \right\rangle d\tilde{\gamma}(x, y)$$

$$\leq \frac{K_{1d}}{\lambda} C_2^2(\tilde{\gamma})$$

where the last line was obtained using the Cauchy-Schwarz inequality on $\mathcal{H}$, RKHS norm homogeneity, the $\frac{1}{\lambda}$-bound on $\|(L + \lambda I)^{-1}\|$, and then the Cauchy-Schwarz inequality on $\mathbb{R}^d$. Using now Taylor's inequality upper bounding the second derivative of $g$ between $t = 0$ and $t = 1$, we have that:

$$g(1) - g(0) \geq \int (y - x)^\top \nabla h_0^\star(x) \mathrm{d}\tilde{\gamma}(x, y) + \mathcal{O}(C_2^2(\tilde{\gamma})).$$

Since $\mathcal{O}(C_2^2(\tilde{\gamma})) = o(C_2(\tilde{\gamma}))$, it follows that $(I \times (1 + \lambda) \nabla h_0^\star)_\# \mathbb{P}_0 \in \partial \mathrm{KALE}(\mathbb{P}_0 \,||\, \mathbb{Q})$.

To prove the reverse implication, assume $v(\overset{\Delta}{=} (1 + \lambda)\tilde{v}) \neq (1 + \lambda)\nabla h_0^\star$. Fix $u > 0$, and choose an "adversarial" $\mathbb{P}_{1,u}$ defined as $\mathbb{P}_{1,u} = (x \longmapsto x + u(1 + \lambda)(\tilde{v}(x) - \nabla h_0^\star(x)))_\# \mathbb{P}_0$, with an associated coupling $\tilde{\gamma} = (x \times (x \longmapsto x + (1 + \lambda)u(\tilde{v}(x) - \nabla h_0^\star(x))))_\# \mathbb{P}_0$. We then have, using a Taylor inequality *lower bounding* the second derivative of $g$:

$$g(1) - g(0) - \int (y - x)^\top (1 + \lambda)\tilde{v}(x) d\tilde{\gamma}(x, y) \leq \int (y - x)^\top (1 + \lambda)(\nabla h_0^\star(x) - \tilde{v}(x)) d\tilde{\gamma}(x, y)$$

$$+ \mathcal{O}(C_2^2(\tilde{\gamma}))$$

$$\leq -u(1 + \lambda) \int \|\tilde{v}(x) - \nabla h_0^\star(x)\|^2 \, \mathrm{d}\mathbb{P}_0(x)$$

$$+ \mathcal{O}(C_2^2(\tilde{\gamma})).$$

In the limit $\mathbb{P}_{1,u} \rightharpoonup \mathbb{P}_0$, e.g. $u \to 0$), the right-hand side scales in $u$, which is the same scaling as $C_2(\tilde{\gamma}) = (\int \|x_1 - x_2\|^2 \, d\tilde{\gamma}(x_1, x_2))^{1/2} = u(1 + \lambda)(\int \|\tilde{v}(x) - \nabla h_0^\star(x)\|^2 \, d\mathbb{P}_0(x))^{1/2}$. Thus, it follows that the inequality:

$$g(1) - g(0) - \int (y - x)^\top (1 + \lambda)\tilde{v}(x) d\tilde{\gamma}(x, y) \geq o(C_2(\tilde{\gamma}))$$

cannot be verified unless $\tilde{v} = \nabla h_0^\star$, $\mathbb{P}_0$a-e. $\qquad \square$

We are now ready to make the following claim:

**Proposition 7** (KALE's gradient flow). *The Wasserstein-2 KALE's gradient flow of KALE on $\mathcal{P}_2(\mathbb{R}^d)$ follows:*

$$\partial_t \mathbb{P}_t - div\left(\mathbb{P}_t \nabla \frac{\delta KALE}{\partial \mathbb{P}}\right) = 0$$

*Proof.* This is a direct application of [3, Definition 11.1.1] using the expression of KALE's strong subdifferential of the form $(i \times v)_\# \mathbb{P}$. $\qquad \square$

Now that we identified the expression of the KALE gradient flow, we will show that the KALE gradient flow admits a unique solution. To prove that the KALE gradient flow admits a unique solution is to prove that KALE is $-M$-semiconvex, for some $M > 0$.

**Lemma 4.** $\mathbb{P} \longmapsto KALE(\mathbb{P} \,||\, \mathbb{Q})$ *is* $-\frac{K_1 d + 4\sqrt{K K_2 d}}{\lambda}$-*geodesically convex.*

*Proof.* Let $\mathbb{P}_a, \mathbb{P}_b \in \mathcal{P}_2(\mathbb{R}^d)$, and consider an admissible coupling $\gamma \in \Gamma(\mathbb{P}_a, \mathbb{P}_b)$ with associated transport costs (for various $p$) $C_p(\gamma) = (\int \|x - y\|^p \, \mathrm{d}\gamma(x, y))^{\frac{1}{p}}$. We consider $(\mathbb{P}_t)_{0 \leq t \leq 1}$ (where $\mathbb{P}_t = (t\pi^2 + (1 - t)\pi^1)_\# \gamma$) a constant-speed geodesic between $\mathbb{P}_a$ and $\mathbb{P}_b$. To prove the geodesic convexity of the KALE, we follow a similar approach as in [18] (Lemma B.2). In particular, we show that $t \longmapsto g(t) = \mathrm{KALE}(\mathbb{P}_t \,||\, \mathbb{Q})$ has an $MC_2^2(\gamma)$-Lipschitz derivative, with some $M$ to be determined. Using a similar implicit function theorem argument as in the proof of Lemma 2, we have:

$$g'(t) = \int (x - y)^\top \nabla h_t^\star(ty + (1 - t)x) d\gamma(x, y).$$

Given $t_1, t_2$, we thus have:

$$|g'(t_1) - g'(t_2)| \leq (I) + (II),$$

where:

$$(I) = \left| \int (x-y)^\top \left( \nabla h^\star_{t_1}(t_1 y + (1-t_1)x) - \nabla h^\star_{t_1}(t_2 y + (1-t_2)x) \right) d\gamma(x,y) \right|$$

$$\leq \left\| h^\star_{t_1} \right\| \sqrt{KK_{2d}}(t_2 - t_1) \int \|x-y\|^2 \, \mathrm{d}\gamma(x,y) \leq \frac{4\sqrt{KK_{2d}}}{\lambda}(t_1 - t_2)C_2^2(\gamma)$$

and:

$$(II) = \int (x-y)^\top \left( \nabla h^\star_{t_1}(t_2 y + (1-t_2)x) - \nabla h^\star_{t_2}(t_2 y + (1-t_2)x) \right) d\gamma(x,y)$$

$$= \int \sum_{i=1}^d (x_i - y_i) \left\langle h^\star_{t_1} - h^\star_{t_2}, \frac{\partial k_{t_2 y + (1-t_2)x}}{\partial x_i} \right\rangle d\gamma(x,y)$$

$$\overset{(i)}{\leq} \int \left\| h^\star_{t_1} - h^\star_{t_2} \right\| \sum_{i=1}^d |x_i - y_i| \left\| \frac{\partial k_{t_2 y + (1-t_2)x}}{\partial x_i} \right\| d\gamma(x,y)$$

$$\overset{(ii)}{\leq} \sqrt{K_{1d}} \left\| h^\star_{t_1} - h^\star_{t_2} \right\| \int \sqrt{\|x-y\|}^2 d\gamma(x,y)$$

$$\overset{(iii)}{\leq} \frac{(t_2 - t_1)K_{1d}C_2^2(\mathbb{P}_a, \mathbb{P}_b)}{\lambda}$$

where (i) follows from Cauchy-Schwarz on $\mathcal{H}$, (ii) uses Cauchy-Schwarz on $\mathbb{R}^d$ and (iii) relies on Lemma 8 and Jensen inequality. We thus conclude that $g'(t)$ is $MC_2^2(\gamma)$-Lipschitz, with $M = \frac{K_{1d}+4\sqrt{KK_{2d}}}{\lambda}$, and thus that KALE is $-M$-geodesically semiconvex $\qquad \square$

The geodesic convexity of the KALE allows to conclude the proof of Proposition 2: indeed, since the KALE is geodesically semiconvex in $\mathbb{P}$, and admits strong extended Fréchet subdifferentials, we conclude that the KALE gradient flow solutions exist and are unique, as guaranteed by [3, Theorem 11.2.1]. $\qquad \square$

# E   Proof of Proposition 3

We recall the following definitions: given a positive measure $\mathbb{P}$, and a function $f \in \mathcal{C}^1(\mathbb{R}^d)$, the weighted Sobolev *semi-norm* of $f$ is given by:

$$\|f\|_{\dot{H}(\mathbb{P})} = \left( \int \|\nabla f\|^2 \, \mathrm{d}\mathbb{P} \right)^{\frac{1}{2}}.$$

Note the important role of the weighted Sobolev semi-norm in the energy dissipation formula of KALE's gradient flow:

$$\frac{\mathrm{dKALE}(\mathbb{P}_t \, || \, \mathbb{Q})}{\mathrm{d}t} = -\int (1+\lambda)^2 \, \|\nabla h^\star\|^2 \, \mathrm{d}\mathbb{P}_t = -(1+\lambda)^2 \, \|h^\star\|^2_{\dot{H}(\mathbb{P}_t)}. \tag{29}$$

By duality, one can define the (possibly infinite) negative *weighted negative Sobolev distance* [4] between $\mu$ and $\nu$:

$$\|\mu - \nu\|_{\dot{H}^{-1}(\mathbb{P})} = \sup_{\|f\|_{\dot{H}(\mathbb{P})} \leq 1} \left| \int f \mathrm{d}(\mu - \nu) \right|.$$

As proven in [50], the weighted negative Sobolev distance linearizes the Wasserstein distance, and one can formally write:

$$W_2(\mu, \mu + d\mu) = \|d\mu\|_{\dot{H}^{-1}(\mathbb{P})} + o(d\mu).$$

Moreover, for all $f \in \mathcal{C}^1(\mathbb{R}^d)$, and $\mu \in \mathcal{M}(\mathbb{R}^d)$, one has:

$$\int f \mathrm{d}\mu \leq \|f\|_{\dot{H}(\mathbb{P})} \|\mu\|_{\dot{H}^{-1}(\mathbb{P})}. \tag{30}$$

To prove Proposition 3, we use the $\lambda$-strong concavity of $\mathcal{K}(h, \mathbb{P})$ w.r.t. $h$ :

$$\text{KALE}(\mathbb{P} \mid\mid \mathbb{Q}) = (1+\lambda)\mathcal{K}(h^\star, \mathbb{P}) \leq (1+\lambda)(\mathcal{K}(0, \mathbb{P}) + \langle h^\star, \nabla_h \mathcal{K}(0, \mathbb{P}) \rangle - \frac{\lambda}{2} \|h^\star\|^2)$$

$$\leq (1+\lambda) \langle h^\star, \mu_{\mathbb{P}} - \mu_{\mathbb{Q}} \rangle = (1+\lambda) \int h^\star(x) \mathrm{d}\mathbb{P} - \int h^\star(x) \mathrm{d}\mathbb{Q}$$

$$\leq (1+\lambda) \|h\|_{\dot{H}(\mathbb{P})} \|\mathbb{P} - \mathbb{Q}\|_{\dot{H}^{-1}(\mathbb{P})} \leq (1+\lambda)C \|h\|_{\dot{H}(\mathbb{P})} .$$

Here we successively applied Eq. (30) and the hypothesis $\|\mathbb{P} - \mathbb{Q}\|_{\dot{H}(\mathbb{P})} \leq C$. Recalling Eq. (29), one has:

$$\frac{d\text{KALE}(\mathbb{P}_t, \mathbb{Q})}{\mathrm{d}t} \leq -\frac{\text{KALE}(\mathbb{P}_t \mid\mid \mathbb{Q})^2}{C^2} \implies \frac{d(1/\text{KALE}(\mathbb{P}_t, \mathbb{Q}))}{\mathrm{d}t} \geq \frac{1}{C},$$

from which the desired inequality follows. $\qquad\square$

**Proof of Proposition 5** We rely on the proof technique used in [4, E.1]. From Lemma 7, we get that assumptions A, D of [4] hold with $L = \sqrt{KK_{2d}}$ and $\lambda^2 = K_{2d}$. Moreover, we know from Lemma 5 that $h^\star$ is $\frac{4K}{\lambda}$-Lipschitz. From these smoothness conditions, all steps in [4, E.1], follow until:

$$\text{KALE}(\mathbb{P}_{n+1} \mid\mid \mathbb{Q}) - \text{KALE}(\mathbb{P}_n \mid\mid \mathbb{Q}) \leq -\gamma \left(1 - \frac{3}{2}\gamma\sqrt{KK_{2d}}\right) \mathcal{D}_{\beta_n}(\mathbb{P}_n) + \gamma\sqrt{K_{2d}}\beta_n \|h^\star\| \mathcal{D}_{\beta_n}(\mathbb{P}_n)^{\frac{1}{2}}.$$

Now, given that $\|h^\star\|^2 \leq \frac{2\text{KALE}(\mathbb{P}_n, \mathbb{Q})}{\lambda}$ and that $\frac{8K_{2d}\beta_n^2}{\lambda^2}\text{KALE}(\mathbb{P}_n, \mathbb{Q}) \leq \mathcal{D}_{\beta_n}(\mathbb{P}_n)$ we have:

$$\text{KALE}(\mathbb{P}_{n+1} \mid\mid \mathbb{Q}) - \text{KALE}(\mathbb{P}_n \mid\mid \mathbb{Q}) \leq -\gamma \left(1 - \frac{3}{2}\gamma\sqrt{KK_{2d}}\right) \mathcal{D}_{\beta_n}(\mathbb{P}_n) + \gamma\sqrt{\frac{2}{8}}D_{\beta_n}(\mathbb{P}_n)$$

$$\leq -\frac{\gamma}{2}\left(1 - 3\gamma\sqrt{KK_{2d}}\right) \mathcal{D}_{\beta_n}(\mathbb{P}_n)$$

$$\overset{(iv)}{\leq} -4\gamma\left(1 - 3\gamma\sqrt{KK_{2d}}\right) \frac{K_{2d}}{\lambda^2}\beta_n^2\text{KALE}(\mathbb{P} \mid\mid \mathbb{Q})$$

$$\overset{(v)}{\leq} -\Gamma\beta_n^2\text{KALE}(\mathbb{P}_n \mid\mid \mathbb{Q}),$$

where (iv) uses the noise schedule assumption and in (v) we noted $\Gamma = 4\gamma\left(1 - 3\gamma\sqrt{KK_{2d}}\right)\frac{K_{2d}}{\lambda^2}$, and the result follows as in [4].

## F   Proof of Proposition 4

We recall the update equations defining the trajectories $(Y_n^{(i)})_{n \leq n_{\max}}$ and $(\bar{Y}_n^{(i)})_{n \leq n_{\max}}$ :

$$Y_{n+1}^{(i)} = Y_n^{(i)} - \gamma(1+\lambda)\nabla\widehat{h}_n^\star(Y_n^{(i)}),$$
$$\bar{Y}_{n+1}^{(i)} = \bar{Y}_n^{(i)} - \gamma(1+\lambda)\nabla h_n^\star(\bar{Y}_n^{(i)}). \tag{31}$$

We denote $c_n = \sqrt{\frac{1}{N}\sum_{i=1}^{N}\mathbb{E}\left\|\bar{Y}_n^{(i)} - Y_n^{(i)}\right\|^2}$. Note that

$$\mathbb{E}W_2(\overline{\mathbb{P}}_n^N, \widehat{\mathbb{P}}_n^N)^2 \leq \frac{1}{N}\sum_{i=1}^{N}\mathbb{E}\left[\left\|Y_{n+1}^{(i)} - \bar{Y}_{n+1}^{(i)}\right\|^2\right] = c_n^2.$$

The iterates $c_n$ satisfy the following recursion:

$$c_{n+1} = \sqrt{\frac{1}{N}\sum_{i=1}^{N}\mathbb{E}\left[\left\|Y_{n+1}^{(i)} - \bar{Y}_{n+1}^{(i)}\right\|^2\right]}$$

$$\leq \sqrt{\frac{1}{N}\sum_{i=1}^{N}\mathbb{E}\left[\left\|Y_n^{(i)} - \bar{Y}_n^{(i)} - \gamma(1+\lambda)\left(\nabla\widehat{h}_n^{\star}(Y_n^{(i)}) - \nabla h_n^{\star}(\bar{Y}_n^{(i)})\right)\right\|^2\right]}$$

$$\leq c_n + \underbrace{\frac{\gamma(1+\lambda)}{\sqrt{N}}\sqrt{\sum_{i=1}^{N}\mathbb{E}\left[\left\|\nabla\widehat{h}_n^{\star}(Y_n^{(i)}) - \nabla h_n^{\star}(\bar{Y}_n^{(i)})\right\|^2\right]}}_{\triangleq A}.$$

Using a triangular inequality, we now split (A) into terms that will be handled differently:

$$c_{n+1} \leq c_n + \gamma(1+\lambda)\left(\underbrace{\frac{1}{\sqrt{N}}\sqrt{\sum_{i=1}^{N}\mathbb{E}\left[\left\|\nabla\widehat{h}_n^{\star}(Y_n^{(i)}) - \nabla\widehat{h}_n^{\star}(\bar{Y}_n^{(i)})\right\|^2\right]}}_{(i)}\right.$$

$$\left. + \underbrace{\frac{1}{\sqrt{N}}\sqrt{\sum_{i=1}^{N}\mathbb{E}\left[\left\|\nabla\widehat{h}_n^{\star}(\bar{Y}_n^{(i)}) - \nabla\bar{h}_n^{\star}(\bar{Y}_n^{(i)})\right\|^2\right]}}_{(ii)} + \underbrace{\frac{1}{\sqrt{N}}\sqrt{\sum_{i=1}^{N}\mathbb{E}\left[\left\|\nabla\bar{h}_n^{\star}(\bar{Y}_n^{(i)}) - \nabla h_n^{\star}(\bar{Y}_n^{(i)})\right\|^2\right]}}_{(iii)}\right)$$

Where we introduced the notation $\bar{h}_n^{\star} = \arg\max_h \mathcal{K}(h, \overline{\mathbb{P}}_n^N)$, the witness function that estimates the *true* witness function $h_n^{\star}$ using $\overline{\mathbb{P}}_n^N$, the empirical version of $\mathbb{P}_n$, instead of $\mathbb{P}_n$. Let us explain the source of each of the terms in the last inequality:

- (i) comes from evaluating the velocity field $\widehat{h}_n^{\star}$ at different points $Y_n^{(i)}$ and $\bar{Y}_n^{(i)}$,
- (ii) comes from using *biased* samples $\{Y_n^{(i)}\}_{i=1}^{N}$ to compute $\widehat{h}_n^{\star}$, and unbiased samples $\{\bar{Y}_n^{(i)}\}_{i=1}^{N}$ to compute $\bar{h}_n^{\star}$.
- (iii) comes from the use of a finite number of unbiased samples to compute $\bar{h}_n^{\star}$.

After controlling (i), (ii), (iii), as detailed below, we get the following upper bound:

$$c_{n+1} \leq c_n\gamma(1+\lambda)\left(1 + \frac{4\sqrt{KK_{2d}} + K_{2d}}{\lambda}\right) + \frac{\gamma(1+\lambda)}{\lambda}\sqrt{\frac{KK_{2d}(1 + e^{\frac{8K}{\lambda}})}{N}}.$$

We use [4, Lemma 26] to conclude:

$$c_n = \sqrt{\frac{2KK_{1d}(1 + e^{\frac{8K}{\lambda}})}{N}} \times \frac{1}{4\sqrt{KK_{1d}} + K_{2d}}(e^{\gamma(1+\lambda)\frac{4\sqrt{KK_{1d}} + K_{2d}}{\lambda}n} - 1).$$

The result on $\mathbb{E}W_2(\bar{\mathbb{P}}_n, \widehat{\mathbb{P}}_n)$ follows by noting that $\mathbb{E}W_2(\bar{\mathbb{P}}_n^N, \widehat{\mathbb{P}}_n^N) \leq \sqrt{\mathbb{E}W_2^2(\bar{\mathbb{P}}_n^N, \widehat{\mathbb{P}}_n^N)}$ by Jensen's inequality. $\qquad\square$

### F.1  Control of the 3 error terms

**Controlling (i)**  To control the first term, we rely on the RKHS derivative reproducing property [67]: $\frac{\partial h}{\partial x_i} = \langle\partial_i k_x, h\rangle$, Assumption 2, and on the uniform bound on $\|h^{\star}\|$ (for all $\mathbb{P}$, $\mathbb{Q}$) given by

(Lemma 5) :

$$\left\|\nabla\widehat{h}_n^\star(Y_n^{(i)}) - \nabla\widehat{h}_n^\star(\bar{Y}_n^{(i)})\right\|^2 \leq \sum_{i=1}^d \left\|\partial_i k_{Y_n^{(i)}} - \partial_i k_{\bar{Y}_n^{(i)}}\right\|^2 \left\|\widehat{h}_n\right\|^2 = \frac{16KK_{2d}}{\lambda^2}\left\|Y_n^{(i)} - \bar{Y}_i^{(n)}\right\|^2.$$

Consequently, we have

$$(i) = \frac{1}{\sqrt{N}}\sqrt{\sum_{i=1}^N \mathbb{E}\left\|\nabla\widehat{h}_n^\star(Y_n^{(i)}) - \nabla\widehat{h}_n^\star(\bar{Y}_n^{(i)})\right\|^2} \leq \frac{4\sqrt{KK_{2d}}}{\lambda\sqrt{N}}c_n.$$

**Controlling (ii)** To control $(ii)$, we rely on Lemma 8, that guarantees that KALE($\mathbb{P} \parallel \mathbb{Q}$) is $\frac{\sqrt{K_{1d}}}{\lambda}$-Lipschitz in $\mathbb{P}$ and $\mathbb{Q}$, when $\mathcal{P}(\mathbb{R}^d)$ is endowed with the Wasserstein-2 metric:

$$\left\|\nabla\widehat{h}_n^\star(\bar{Y}_n^{(i)}) - \nabla\bar{h}_n^\star(\bar{Y}_n^{(i)})\right\|^2 = \sum_{j=1}^d \left(\partial_j\widehat{h}_n^\star(\bar{Y}_n^{(i)}) - \partial_j\bar{h}_n^\star(\bar{Y}_n^{(i)})\right)^2$$

$$\leq K_{1d}\left\|\widehat{h}_n^\star - \bar{h}_n^\star\right\|^2.$$

Consequently, using Lemma 8, we have:

$$\left\|\nabla\widehat{h}_n^\star(\bar{Y}_n^{(i)}) - \nabla\bar{h}_n^\star(\bar{Y}_n^{(i)})\right\|^2 \leq \frac{K_{1d}^2}{\lambda^2}W_2(\widehat{\mathbb{P}}_n^N, \bar{\mathbb{P}}_n^N)^2$$

$$\implies (ii) = \frac{1}{\sqrt{N}}\sqrt{\sum_{i=1}^N \mathbb{E}\left\|\nabla\widehat{h}^\star(\bar{Y}_n^{(i)}) - \nabla h^\star(\bar{Y}_n^{(i)})\right\|^2} \leq \frac{K_{1d}}{\lambda\sqrt{N}}\sqrt{\mathbb{E}W_2^2(\widehat{\mathbb{P}}_n^N, \bar{\mathbb{P}}_n^N)} \leq \frac{K_{1d}}{\lambda\sqrt{N}}c_n.$$

**Controlling (iii)** In (iii), the witness function $\bar{h}_n^\star$ is an empirical version of $h_n^\star$. Repeating the first lines of (ii), we have:

$$\left\|\nabla\bar{h}_n^\star(\bar{x}_n^{(i)}) - \nabla h_n^\star(\bar{x}_n^{(i)})\right\|^2 \leq K_{1d}\left\|\bar{h}_n^\star - h_n^\star\right\|^2.$$

We could use the bound given in $(ii)$ to get a bound on $\left\|\bar{h}_n^\star - h_n^\star\right\|$, but the sample complexity of the Wasserstein distances scales in $\mathcal{O}(n^{-1/d})$, which is much slower than our target rate $1/\sqrt{N}$ [65]. Instead, we rely on the concentration inequality given by Lemma 6, ensuring that $\mathbb{E}\left\|\bar{h}_n^\star - h_n^\star\right\|^2 \leq \frac{2K(1+e^{\frac{8K}{\lambda}})}{N\lambda^2}$. Following this, we have:

$$(iii) = \frac{1}{\sqrt{N}}\sqrt{\sum_{i=1}^N \mathbb{E}\left\|\nabla\bar{h}_n^\star(\bar{x}_n^{(i)}) - \nabla h^\star(\bar{x}_n^{(i)})\right\|^2} \leq \frac{1}{\lambda}\sqrt{\frac{2KK_{1d}(1+e^{\frac{8K}{\lambda}})}{N}}.$$

# G  Auxiliary Lemmas

**Lemma 5** (Uniform smoothness of the KALE witness function)**.** *Under Assumption 1, and for all $\mathbb{P}$, $\mathbb{Q}$, the following inequalities hold:*

$$\frac{\lambda}{2}\|h^\star\|^2 \leq KALE(\mathbb{P} \parallel \mathbb{Q}) \leq 2\sqrt{K}\|h^\star\|,$$

*implying* $\|h^\star\| \leq \frac{4\sqrt{K}}{\lambda}$. *We also have the finer estimate* $\|h^\star\| \leq \frac{2MMD(\mathbb{P}\|\mathbb{Q})}{\lambda}$.

*Proof.* The right inequality follows from the proof of Proposition 3. Indeed, we have:

$$KALE(\mathbb{P} \parallel \mathbb{Q}) \leq \langle h^\star, \mu_\mathbb{P} - \mu_\mathbb{Q}\rangle \leq \|h^\star\|(\|\mu_\mathbb{P}\| + \|\mu_\mathbb{Q}\|) \leq 2\sqrt{K}\|h^\star\|.$$

The left inequality can be noticed using KALE's *dual formulation* Eq. (6)

$$\text{KALE}(\mathbb{P} \,||\, \mathbb{Q}) = \underbrace{\int \left( f^\star (\log f^\star - 1) + 1 \right) \mathrm{d}\mathbb{Q}}_{\geq 0} + \frac{1}{2\lambda} \left\| \int f^\star(x)\mathrm{d}\mathbb{Q}(x) - \mu_\mathbb{P} \right\|^2 \tag{32}$$

$$\geq \frac{1}{2\lambda} \left\| \int f^\star(x)\mathrm{d}\mathbb{Q}(x) - \mu_\mathbb{P} \right\|^2 = \frac{\lambda}{2} \|h^\star\|^2 .$$

To get the finer estimate, we keep track of $\frac{\lambda}{2}\|h^\star\|_{\mathcal{H}}^2$ term. By convexity of $\exp$, we have:

$$\underbrace{1 + \int h\mathrm{d}\mathbb{P} - \int e^h \mathrm{d}\mathbb{Q} - \frac{\lambda}{2}\|h\|^2}_{\mathcal{K}(h,\mathbb{P})} \leq \int h\mathrm{d}\mathbb{P} - \int h\mathrm{d}\mathbb{Q} - \frac{\lambda}{2}\|h\|^2 .$$

Recalling now that $\mathcal{K}(h^\star, \mathbb{P}) \geq \mathcal{K}(0, \mathbb{P}) = 0$, we must have:

$$\int h^\star \mathrm{d}\mathbb{P} - \int h^\star \mathrm{d}\mathbb{Q} - \frac{\lambda}{2}\|h^\star\|^2 \geq 0$$

$$\implies \|h^\star\| \leq \frac{2\|f_{\mathbb{P},\mathbb{Q}}\|}{\lambda}$$

Where the last line used the Cauchy-Schwarz inequality. □

**Lemma 6.** *Under Assumption 1, and using the notations of Appendix F, we have:*

$$\mathbb{E}\left\|\bar{h}_n^\star - h_n^\star\right\|^2 \leq \frac{2K(1 + e^{\frac{8K}{\lambda}})}{N\lambda^2}$$

*Proof.* We first notice, as explained in [6] (Proposition 12), that $\left\|\bar{h}_n^\star - h_n^\star\right\| \leq \frac{1}{\lambda}\left\|\nabla\widehat{\mathcal{L}}(h_n^\star) - \nabla\mathcal{L}(h_n^\star)\right\|$ where $\mathcal{L} = 1 + \int h\mathrm{d}\mathbb{P} - \int e^h\mathrm{d}\mathbb{Q}$ is the KL objective, and $\widehat{\mathcal{L}}(h) = \int h\mathrm{d}\mathbb{P}_n^N - \int e^h\mathrm{d}\widehat{\mathbb{Q}}^N + 1$ is its empirical equivalent. We then use [61] (Proposition A.1, notice that their statement also holds for $(\mathbb{E}\|\int r\mathrm{d}\mathbb{P}_n - \int r\mathrm{d}\mathbb{P}\|^2)^{1/2}$), to get:

$$E\left\|\nabla\widehat{\mathcal{L}}(h_n^\star) - \nabla\mathcal{L}(h_n^\star)\right\|^2 \leq \mathbb{E}\left\|\int k(x,\cdot)\mathrm{d}\mathbb{P}_n - \int k(x,\cdot)d\bar{\mathbb{P}}_n^N\right\|^2$$

$$+ \mathbb{E}\left\|\int k(x,\cdot)e^{h_n^\star}\mathrm{d}\mathbb{Q} - \int k(x,\cdot)e^{h_n^\star}\mathrm{d}\widehat{\mathbb{Q}}^N\right\|^2$$

$$\leq \frac{K(1 + e^{\frac{8K}{\lambda}})}{N},$$

where we used the Cauchy-Schwarz inequality on $\mathcal{H}$ and Lemma 5 to bound the squared norm of $x \longmapsto k(x,\cdot)e^{h^\star(x)}$. □

**Lemma 7.** *Under Assumption 2, The maps $x \longmapsto k_x(\stackrel{\Delta}{=} k(x,\cdot))$ and $x \longmapsto \nabla k_x$ are differentiable. Moreover, we have*

$$\|k_x - k_y\| \leq \sqrt{K_{1d}}\|x - y\|$$
$$\|\nabla k_x - \nabla k_y\| \leq \sqrt{K_{2d}}\|x - y\|$$

*Proof.* We prove the differentiability and the Lispchitzness property for the map $x \longmapsto k_x$; the arguments can be straightforwardly adapted to the case of $x \longmapsto \nabla k_x$. To prove the differentiability, we build upon [60, Lemma 4.34], that guarantees that $x \longmapsto k(x,\cdot)$ admits partial derivatives for all $i$, noted $\partial_i\phi(x)$. We finish the proof by construction: let $D\phi(x) : \mathbb{R}^d \longmapsto \mathcal{H}$ our candidate differential, defined as $D\phi(x)(\Delta) = \sum_{i=1}^d \Delta_i \partial_i\phi(x)$ for all $\Delta \in \mathbb{R}^d$. We show that $D\phi(x)$ is the differential of $\phi$

at x using a simple telescopic argument: let us note $(x + \Delta)_{:i} = (x_1 + \Delta_1, \ldots, x_i + \Delta_i, x_{i+1}, \ldots, x_d)$ for any $i \in \{0, \ldots, d\}$ with $(x + \Delta)_{:0} = x$ by convention. Then:

$$\phi(x + \Delta) - \phi(x) = \sum_{i=d}^{1} \phi(x + \Delta)_{:i} - \phi((x + \Delta)_{:i-1})$$

Knowing that $\phi((x + \Delta)_{:i}) - \phi((x + \Delta)_{:(i-1)}) = \partial_i \phi((x + \Delta)_{:i-1}) \Delta_i + o(|\Delta_i|)$, we have:

$$\phi(x + \Delta) - \phi(x) - D\phi(x)(\Delta) = \sum_{i=1}^{d} (\partial_i \phi((x + \Delta)_{:i-1}) - \partial_i \phi(x)) \Delta_i + o(|\Delta_i|)$$

$$\implies \|\phi(x + \Delta) - \phi(x) - D\phi(x)(\Delta)\| \le \sum_{i=1}^{d} |\Delta_i| \|\partial_i \phi((x + \Delta)_{:i-1}) - \partial_i \phi(x)\| + o(\|\Delta\|_1)$$

From [60, Lemma 4.34], we have: that:
$$\|\partial_i \phi((x + \Delta)_{:i-1}) - \partial_i \phi(x)\|^2 = A - B$$

where
$$A = \partial_i \partial_{i+d} k((x + \Delta)_{:i-1}, (x + \Delta)_{:i-1}) - \partial_i \partial_{i+d} k((x + \Delta)_{:i-1}, x))$$
$$B = \partial_i \partial_{i+d} k((x + \Delta)_{:i-1}, x) - \partial_i \partial_{i+d} k(x, x)$$

Since $\partial_i \partial_{i+d} k(x, x')$ is continuous, both $A$ and $B$ tend to 0 as $\|\Delta\|$ tends to 0. Thus, we have:

$$\|\phi(x + \Delta) - \phi(x) - D\phi(x)(\Delta)\| \le \sum_{i=1}^{d} o(|\Delta_i|) + o(\|\Delta\|_1) = o(\|\Delta\|_2)$$

by equivalency of $\|\cdot\|_1$ and $\|\cdot\|_2$ in $\mathbb{R}^d$.

Lipschitzness is guaranteed by bounding the operator norm of $D\phi(x)$:

$$D\phi(x) = \sup_{\|\Delta\| = 1} \|D\phi(x)\Delta\| \le \sum_{i=1}^{n} |\Delta_i| \|\partial_i \phi(x)\| \le \sqrt{\|\Delta\|^2} \sqrt{\sum_{i=1}^{n} \|\partial_i \phi(x)\|^2} = \sqrt{K_{1d}}$$

$\square$

**Lemma 8.** *For any $\mathbb{P}_0, \mathbb{P}_1 \in \mathcal{P}_2(\mathbb{R}^d)$, with associated KALE witness functions $h_0^\star, h_1^\star$, we have:*
$$\|h_1^\star - h_0^\star\|^2 \le \frac{K_{1d}}{\lambda^2} W_2(\mathbb{P}_0, \mathbb{P}_1)^2.$$

*Proof.* The optimal functions $h_0^\star$ and $h_1^\star$ are characterized by the following optimality condition:
$$\int k(x, \cdot) d\mathbb{P} - \int k(x, \cdot) e^{h^\star} d\mathbb{Q} - \lambda h^\star = 0.$$

Let us now pose $d\mathbb{P}_t = d\mathbb{P}_0 + td\chi$ with $d\chi = d\mathbb{P}_1 - d\mathbb{P}_0$, and its associated witness function $h_t^\star$. Using an implicit function theorem argument [32] between $t$ and $h_t^\star$, we can write, with notations of Appendix D:
$$\frac{dh_t^\star}{dt} = (\boldsymbol{L}(h_t^\star) + \lambda I)^{-1} \int k(x, \cdot)(d\mathbb{P}_1 - d\mathbb{P}_0).$$

The operator $\boldsymbol{L}$ is the covariance operator of the measure $\tilde{\mathbb{Q}} = e^{h^\star} \mathbb{Q}$. This operator is compact given that $k$ is bounded by Assumption 1. Using the spectral theorem on Hilbert spaces, we know that there exists a complete orthonormal system of eigenvectors of $\boldsymbol{L}$, with associated eigenvalues $\{\mu_{i,t}\}_{i \in \mathbb{N}}$ for any $t$. The operator $(\boldsymbol{L} + \lambda I)^{-1}$ admits an identical eigendecomposition, with eigenvalues $\left\{\frac{1}{\lambda + \mu_{i,t}}\right\}_{i \in \mathbb{N}}$: thus, the operator norm of $(\boldsymbol{L} + \lambda I)^{-1}$ is upper-bounded by $1/\lambda$. We can thus extract a bound on $\|h_1^\star - h_0^\star\|^2$:

$$\|h_1^\star - h_0^\star\|^2 = \left\| \int_0^1 (\boldsymbol{L}(h_t^\star) + \lambda I)^{-1} \left( \int k(x, \cdot)(d\mathbb{P}_1 - d\mathbb{P}_0) \right) dt \right\|^2$$

$$\le \int_0^1 \left\| (L(h_t^\star) + \lambda I) \int k(x, \cdot)(d\mathbb{P}_1 - d\mathbb{P}_0) \right\|^2 dt$$

$$\le \int_0^1 \frac{1}{\lambda^2} \left\| \int k(x, \cdot)(d\mathbb{P}_1 - d\mathbb{P}_0) \right\|^2 dt = \frac{1}{\lambda^2} \left\| \int k(x, \cdot)(d\mathbb{P}_1 - d\mathbb{P}_0) \right\|^2.$$

Now, let $\nu \in \Gamma(\mathbb{P}_1, \mathbb{P}_0)$. Then one has:

$$\int k(x, \cdot)\,(\mathrm{d}\mathbb{P}_0 - \mathrm{d}\mathbb{P}_1) = \int \left( k(x, \cdot) - k(y, \cdot) \right) d\nu(x, y)$$

$$\left\| \int k(x, \cdot)\,(\mathrm{d}\mathbb{P}_0 - \mathrm{d}\mathbb{P}_1) \right\|^2 \le \int \| k(x, \cdot) - k(y, \cdot) \|^2 \, \mathrm{d}\nu(x, y)$$

$$\le K_{1d} \int \| x - y \|^2 \, \mathrm{d}\nu(x, y) = K_{1d} W_2(\mathbb{P}_0, \mathbb{P}_1)^2$$

Where we applied first Jensen's inequality and Lemma 7. $\qquad\square$

## H  Details of Numerical Experiments and Impact of Noise Injection

In this section, we provide further details on the experiments in the main paper. The step size $\gamma$ used for the KALE particle descent algorithm scales with $\lambda$ as $\min(0.1, \frac{\lambda}{10})$. For all experiments, we used a Gaussian kernel $k(x, y) = \exp(-\frac{\|x-y\|^2}{2\sigma^2})$. The kernel width $\sigma$ is described for each experiment set.

**"Three rings" experiments**   For this experiment, the number of particles in each distribution was $N = 300$, and we used the Newton algorithm to compute the KALE. We used a kernel width $\sigma = 0.3$. We show in Fig. 4a the impact of noise injection with a constant noise schedule of $\beta_n = 0.3$.

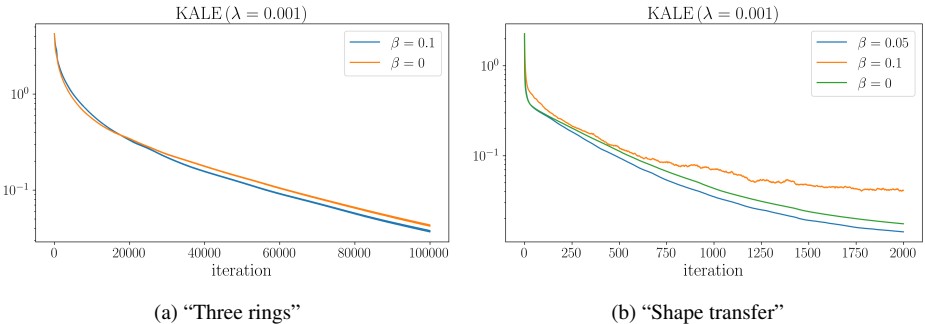

(a) "Three rings"    (b) "Shape transfer"

Figure 4: Impact of noise injection on the KALE value during a KALE particle descent algorithm.

**"Shape transfer" experiments** For this experiment, we used artificial data from the same source as [44]. We sub-sampled both shapes to $N = 2000$ points, and used a kernel width of $\sigma = 0.3$, as well as $\lambda = 0.001$. Because the number of particles is higher in that case, we used a coordinate descent algorithm to compute KALE, that has a complexity in $\mathcal{O}(N^2)$. We show in Fig. 4b the impact of noise injection with a constant noise schedule of $\beta_n = 0.05$. For this experiment, we also show empirically that while using a small amount of noise *lowers* the final KALE value when compared to the unregularized KALE flow, a too large noise level $\beta_n = 0.1$ results in a *larger* final KALE value. We hypothesize that that noise schedule did not respect the assumptions made in Proposition 5.

**"Mixture of Gaussians" experiments** For this experiment, we used $N = 240$ particles for each distribution, and a standard deviation of $0.25$ for each target Gaussian. We used the Unadjusted Langevin Algorithm [23] to simulate a KL gradient flow with step size $0.001$, and the MMD particle descent algorithm of [4] to simulate a MMD flow with step size $0.001$. For both the MMD and the KALE, we used the same Gaussian kernel with kernel width $\sigma = 0.35$. We show the impact of noise injection for the KALE flow with a constant noise schedule $\beta_n = 0.3$ to regularize KALE flow with $\lambda = 0.001, 0.1$ and $10000$.

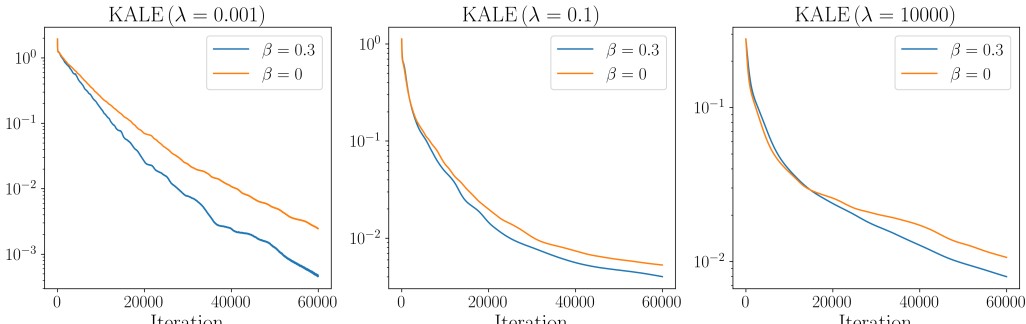

Figure 5: Impact of noise injection: Mixture of Gaussians experiments