# OpenReview forum: "KALE Flow: A Relaxed KL Gradient Flow for Probabilities with Disjoint Support"
_NeurIPS.cc/2021/Conference — NeurIPS 2021 Poster_

### Official Review · Reviewer_Go8M · 2021-07-01

**Rating:** 6
**Confidence:** 3

**Summary:**

This paper proposes a new probability divergence called KALE (KL approximate lower-bound estimator) that interpolates between the KL divergence and MMD through a scaling parameter. The motivation comes from regularizing the dual formulation of KL, and KALE has been shown to exhibit desirable topological properties. In addition, the Wasserstein gradient flow of KALE is derived and a particle-based descent algorithm is introduced to compute the KALE flow in practice. Consistency and global convergence under noise injection of the proposed algorithm have been shown. Experiments on toy examples have been conducted to demonstrate empirically that the KALE flow indeed interpolates between those of KL and MMD, while having a "non-flat" geometry (compared to MMD) and requiring only sample access to compute (compared to KL).

**Limitations And Societal Impact:**

The authors have not addressed the limitations or potential negative societal impact.

**Main Review:**

The theoretical aspect of the paper is quite solid. While the established properties of KALE as a probability divergence are not surprising - after all it interpolates MMD and KL and hence ought to enjoy the properties of both - the exposition is clear and rigorous. It is perhaps a bit lacking in motivation: what motivates the regularization in (2), and what does the term do intuitively? The choice seems rather arbitrary to me. It would also be nice to have some comparison with Sinkhorn divergence, which also interpolates between MMD and KL: what theoretical advantage does KALE have compared to it? why do we need another interpolating divergence?

Using KALE as a gradient flow is interesting as it circumvents a lot of the problems that KL flow has and it is curious that it converges slower than KL. The particle descent algorithm is natural and not very novel, but the consistency of the descent algorithm is well justified. I'm wondering if it is possible to avoid space discretization (particle-based methods)? For instance, you can parameterize $\mathbb{P}_n$'s using normalizing flow, and then using techniques similar to "Stochastic Optimization for Large-scale Optimal Transport" by Genevay et al. for solving for finding $h_n^\star$ in an RKHS. To me, there is this disconnection where a large part of the theory is based on the RKHS theory but the algorithm has no use of it.

For the experiments, it is nice that the properties of KALE flow are checked empirically, but the setups are simple 2D examples. I will be interested in seeing how KALE would fare in higher dimensional tasks, such as when the density is defined on a submanifold. In general, I found the experiments section a bit weak and lacks applications: for what practical problems would people use KALE gradient flow for, compared to alternatives? Some comparison with other gradient flow algorithms (such as the SVGD algorithm by Chewi et al.) on a larger scale setting would be nice.

Some minor comments:
- Line 147: what is $h^*$?
- Line 608: "litterature"
- Figure 2: where is MMD result?
- Algorithm 1 could use some captions to point back what "backprop", "log_ratio" refer to in the main text.

**Time Spent Reviewing:**

5

---

> ### Author Response · Authors · 2021-08-10
> **Thank you for your review**
>
> We thank Reviewer Go8M for their review, and their interesting algorithmic follow ups. We discuss them, as well as other remarks/concerns below:
>
> *[The definition of KALE] is perhaps a bit lacking in motivation: what motivates the regularization in (2), and what does the term do intuitively? The choice seems rather arbitrary to me.*
>
> We answer this question in two parts:
>
> (1) Why should we regularize the KL objective?
>
> The reason is that  the *unregularized* KALE objective ($\lambda = 0$)  has a supremum equal to $\text{KL}(\mathbb  P, \mathbb Q)$, and has thus a limited regularity: for disjoint $\mathbb P$ and $\mathbb Q$, the supremum will equal $+\infty$, and is informally  "reached" at an ill-defined log-density ratio $h^\star$ equal to either $+\infty$ on $\text{supp} \mathbb P \setminus \text{supp} \mathbb Q$ and $-\infty$ on $\text{supp} \mathbb Q \setminus \text{supp} \mathbb P$. The addition of a RKHS norm penalty term to the objective allows to endow KALE with a witness function: a relaxation of the log-density ratio that remains well defined even when $\mathbb P$ and $\mathbb Q$ have disjoint support. In that sense, KALE and its witness function are more "regular" than KL and its "witness function" analogue, $\log \text{d}\mathbb P/\text{d}\mathbb Q$.
>
> (2) What motivates the choice of using a RKHS norm penalty as a regularizer inside the KALE objective?
>
> We agree that this choice is not well explained in the paper. we give two justifications below:
>
> (2).1 Using a RKHS norm penalty allows to control the smoothness of the KALE witness function: indeed,
> - Lemma 5 ensures that the KALE witness function has a bounded RKHS norm.
> - The Lipschitz constant of a RKHS function can be controlled by its RKHS norm $|f(x) - f(y)| \leq \sqrt{2L}  \|\| f\|\|_{\mathcal H} \|\| x - y\|\| $ (using the Cauchy-Schwarz inequality and Lemma 7)
>
>
> (2).2 Moreover, RKHS norms are easily computable for many RKHS functions, and allow exact, implementable algorithms such as the KALE particle descent.
>
> We believe that investigating the use of other regularizers might lead to other useful KALE definitions, and see it as an avenue for future follow-up work. Any suggestions on this topic are warmly welcomed.
>
> *It would also be nice to have some comparison with Sinkhorn divergence, which also interpolates between MMD and KL: what theoretical advantage does KALE have compared to it? why do we need another interpolating divergence?*
>
> We are aware of a result showing that Sinkorn divergences interpolate between the Wasserstein distance and the MMD with a specific kernel [20], but were not aware of a result showing that they interpolate between KL and MMD. If the reviewer has a particular work in mind regarding this last point, we would be happy to compare with it.
>
> *Using KALE as a gradient flow is interesting as it circumvents a lot of the problems that KL flow has and it is curious that it converges slower than KL*
>
> As the reviewer says, KALE circumvents some limitations of KL (such as allowing for targets without densities) by restricting the set of dual functions to a smoother class. This comes at the cost of a slower convergence rate for the KALE flow, although an interesting question for future work is whether such rates can be improved.
>
> *The particle descent algorithm is natural and not very novel, but the consistency of the descent algorithm is well justified. I'm wondering if it is possible to avoid space discretization (particle-based methods)? For instance, you can parameterize the log density ratio using normalizing flow, and then using techniques similar to "Stochastic Optimization for Large-scale Optimal Transport" by Genevay et al. for solving for finding $h^\star$ in an RKHS.*
>
> Trying to avoid space discretization is an excellent objective in general, and we thank the reviewer for bringing that point.
> We can think of two different ways of partially avoiding space discretization, with different applications:
>
> 1. No discretization of the source (moving) probability $\mathbb P$. Typical models that flow from a continuous, parameterized source probability to a discretized target are implicit generative models. Our understanding of the reviewer's comment is that if the source probabilities $\mathbb P$ are parametrized, then one can run a discrete-time, continuous-space, parametrized, approximate KALE flow by sampling from the source probabilities $\mathbb P_n$ at each iteration n in order to compute the KALE, and updating $\mathbb P_n$'s parameter using a KALE gradient step. This idea is interesting, and is implemented in Generalized Energy Based Models [5] (GEBMs) that are flexible energy based models trained using a neural KALE gradient flow. In that case, the source probability is parametrized by a base measure and a energy function defined on the support of that base, which are both parameterized by neural networks. Regarding normalizing flows, these were indeed also used as source measures $\mathbb P$ for GEBMs in [5, Table 3].
>
> 2. No discretization of the target $\mathbb Q$. Typical particle descent algorithms using a continuous/parametrized target $\mathbb Q$ are particle variational inference methods, such as Stein Variational Gradient Descent. In that respect, it is possible to avoid space discretization of the target probability by computing the KALE using the Stein kernel associated with the target $\mathbb Q$. Using such a Stein kernel allows for a semi-discrete KALE flow, and could serve as a sampling algorithm to sample from a (possibly unnormalized) density.
>
>
>  As a clarifying remark, the element lacking a continuous space structure in the KALE particle descent that is proposed in our submission are the probability measures $\mathbb P$ and $\mathbb Q$, but not the witness function/log-density ratio $h^\star$, which is a RKHS function with a known formula (upon solving a convex optimization problem).
>
> Finally, we thank the reviewer for pointing us to "Stochastic Optimization for Large-scale Optimal Transport" by Genevay et. al. Our understanding of this paper is that it proposes an online method to compute the Sinkhorn divergence between $\mathbb P$ and $\mathbb Q$ by parameterizing the dual Sinkhorn potentials using a RKHS. In relation to the KALE flow:
> - We believe that it should be possible to compute the KALE in an online manner by carrying out a stochastic optimization of the KALE objective (although the norm penalty term may be complex to handle).
> - However, we are uncertain of how this idea could help us to avoid space discretization. A possible use of this stochastic KALE computation would be an amortized computation of the KALE at each iterate of the KALE flow, possibly using a parameterized source from which it is possible to draw samples. As discussed below, such a model is implemented in [5]
>
> *To me, there is this disconnection where a large part of the theory is based on the RKHS theory but the algorithm has no use of it.*
>
> The KALE particle descent algorithm is a sample-based discretization of the continuous KALE flow. This particle descent algorithm is based on RKHS the same way the KALE (Flow) itself is based on RKHS: through the restriction of the original dual KL formulation search space to a RKHS, and through the addition of a RKHS norm penalty. This reliance on RKHS theory is not lost after discretization in time and space to obtain the KALE particle descent algorithm. Concretely, (Eq. 6), which is the equation used in the KALE particle descent algorithm, contains kernel functions in the norm penalty term, signalling the importance of RKHS in the KALE.
>
> There may be other elements of RKHS theory that we did not leverage and that we could use to obtain other results about the KALE and its flow, and welcome the reviewer's suggestions on this topic.
>
> *For the experiments, it is nice that the properties of KALE flow are checked empirically, but the setups are simple 2D examples. I will be interested in seeing how KALE would fare in higher dimensional tasks, such as when the density is defined on a submanifold. In general, I found the experiments section a bit weak and lacks applications: for what practical problems would people use KALE gradient flow for, compared to alternatives?*
>
> We thank the reviewer for raising these important questions.
>
> Practical reasons for studying  the KALE flow: gradient flows of probability divergences in machine learning are studied as the idealized (nonparametric, continuous time) variants of GAN training dynamics [3]. In particular, Generalized Energy Based Models [5] can be understood as KALE gradient flows, with the RKHS function class replaced by a neural network, and the particles produced by the generator architecture.
>
> Performance of the KALE on high dimensional data: our focus in the present paper is on a theoretical understanding of KALE flow dynamics, however we point out that Generalized Energy Based Models [5] (a “constrained” instance of KALE flow, as seen above) have been trained on subsampled ImageNet and Cifar-10 data, and were shown to have competitive performance, and give realistic samples.
>
> *Some comparison with other gradient flow algorithms (such as the SVGD algorithm by Chewi et al.) on a larger scale setting would be nice.*
>
> SVGD requires knowledge of the target distribution in closed form, up to normalization. By contrast, the KALE particle descent algorithm in our submission is studied in the fully-discrete, sample-based setting (source and target distribution known only up to samples). Thus, the setting of SVGD and of the KALE particle descent algorithm in our present paper are not directly comparable.

---

> > ### Comment · Reviewer_Go8M · 2021-08-15
> > **Response**
> >
> > I greatly appreciate the authors' detailed response and the clarifications, in particular, on how RKHS theory is used in the particle-based algorithms proposed in the paper.
> >
> > After reading other reviews, I decide not to change my ratings. I want to reiterate that I think the theory is solid in this paper, but the motivation, the formulation, and the style are very close to [3] and [5]. The particle descent algorithm is also very natural. More experiments could be useful in justifying the practical benefit of KALE flow especially in higher-dimensional settings (for example, comparing the particle descent algorithm to [5]). Like I mentioned in my comments, addressing the challenges of spatial discretization could be another good future direction.

---

### Official Review · Reviewer_ftSD · 2021-07-14

**Rating:** 6
**Confidence:** 5

**Summary:**

This paper studies the Wasserstein gradient flow of the KALE, a lower bound estimator of the KL divergence. By time discretization, this flow leads to an algorithm to sample from a target distribution for which samples are known (in contrast to the setting where the log density is known). Moreover, this KALE flow performs well in the case when the initial and the target distribution have disjoint supports, a setting in which the KL flow is not defined.

**Ethical Concerns:**

N/A (theoretical work)

**Limitations And Societal Impact:**

Yes

**Main Review:**

Originality: The paper is in two parts.

The first part is about defining the KALE, a relaxation of the KL which is not equal to infinity if the two distributions are singular. The KALE depends on a relaxation parameter and on the choice of a RKHS, and is shown to interpolate between the MMD and the KL (by setting the parameter accordingly). The main mathematical remark here is that the derivative at zero of the objective defining the KALE is the witness function whose norm is the MMD. This is hidden is the appendix, I would suggest to state it in the main paper as a Lemma.
Except the weak continuity of KALE, this part seems to be mainly an adaptation of [5], for the purpose of defining the KALE flow, which is the topic of the second (and main) part of the paper.

The second part is about the KALE flow. The authors first establish the existence and uniqueness of the KALE flow, a task which is non trivial due to the fact the KALE doesn't have the form of a free energy.  Then, under a boundedness assumption of the flow, they show a O(1/t) convergence rate of KALE to zero along the flow. This results seems to be mainly an extension of a result of [3].
I don't think that the convergence of the KL would be faster (as suggested after Prop 3) because KL does not satisfy log Sobolev inequality.

Additionally, the authors propose their main algorithm, obtained as a time discretization of the KALE flow. This algorithm is a provable approximation of the Euler discretization of the KALE flow (which is intractable).

Finally, they propose a modified algorithm by introducing a Gaussian noise at each iteration of the algorithm. This seems similar to a technique used in [3]. This is the least convincing part of the paper because Prop 5 relies on an assumption that cannot be checked (and I cannot find a case where this assumption holds, it can be the case that this assumption is super strong), and empirically this noise injection does not lead to large improvements. Therefore, the introduction of noise is not really justified.

The simulations of the KALE flow are nice and show what the KALE flow brings compare to other flows such as MMD flow of KL flow. The KALE flow improves the convergence in the case where the initial and the target distribution have disjoint supports.

Refs: Various papers have investigated the task of sampling in the case where source and target have disjoint supports, **in the case where the log density is known** in particular Arxiv 1507.02564, 1802.10174, 2006.09270, 2010.16212









Quality and clarity: The paper is rather clear but would deserve more polishing. There are many typos in the appendix, and also some typos in the main paper (l.13, l. 116, l.214, l.284, Eq (13) etc.).
It would be better to move the assumptions in the main paper (and correct the typos in the statement of the assumptions).
The bibliography has not been cleaned before submission. Refs [1] and [2] are almost the same and the authors refers to [2] as [1] sometimes in the appendix. Venues, names etc. have to be harmonized.
Finally, the implementation of Algo 1 should be discussed a bit more (2 more lines would be sufficient).

Significance: Overall, the content is novel and relevant to the community, but the main results seem close to those in [3] and [5].


MINOR:

l. 57: "generalization properties"? I don't understand
l.92: It seems better to write that we work on an open set X endowed with its Borelian sigma-algebra
l.124: Strong convexity is used to prove that KALE is a probability divergence
l.151: Why not using directly the primal formulation of KALE?
l.295: Why are the heart and the spiral supported on disjoint subsets? This is not seen on the figure.
l.306 "the target admits a positive density"




**Time Spent Reviewing:**

7

---

> ### Author Response · Authors · 2021-08-10
> **Thank you for your review**
>
> We thank Reviewer ftSD for their thoughtful suggestions and review. We discuss the major and minor remarks below.
>
> *Finally, they propose a modified algorithm by introducing a Gaussian noise at each iteration of the algorithm. This seems similar to a technique used in [3]. This is the least convincing part of the paper because Prop 5 relies on an assumption that cannot be checked (and I cannot find a case where this assumption holds, it can be the case that this assumption is super strong), and empirically this noise injection does not lead to large improvements. Therefore, the introduction of noise is not really justified.*
>
> We agree with both reservations about noise raised in this comment (strongness of the hypothesis, and not a decisive factor for convergence of the flow in the submission examples). Nonetheless, noise injection is still beneficial to obtain lower KALE values at convergence, a phenomenon already reported in [3] for the MMD flow. Thus, despite the strong requirements to obtain global convergence, we believe that noise injection could be a robust regularization technique for kernel-based divergences, and we hope that reporting its benefits may trigger future theoretical work to better understand its properties.
>
> Nonetheless, you are correct - the algorithm without noise injection performs well, and remains of interest.
>
> *L.57: "generalization properties"? I don't understand*
>
> This term refers to the estimation properties of the KALE w.r.t KL, which we will use instead of "generalization properties" in the updated version of the paper.
>
> *l.92: It seems better to write that we work on an open set X endowed with its Borelian sigma-algebra l.124: Strong convexity is used to prove that KALE is a probability divergence. l.306 "the target admits a positive density"*
>
> We thank the reviewers for these two suggestions, which will be applied to our submission.
>
> *l.151: Why not using directly the primal formulation of KALE?*
>
> Indeed we can also use the primal formulation and apply the representer theorem to obtain a finite-dimensional (primal) optimization problem with a solution equal to KALE. However, the dual KALE problem, as done (only for discrete measures) in [44] translates immediately to (7) for discrete measures, without any intermediate steps.
>
> *l.295: Why are the heart and the spiral supported on disjoint subsets? This is not seen on the figure*.
>
> The clearest element showing support disjointness are the four corners of the spiral, which are not in the support of the heart. The white spaces in between the spiral bands, on the other hand, are partly contained in the support of the heart. We will clarify this.

---

> > ### Comment · Reviewer_ftSD · 2021-08-16
> > **Thanks for the clarifications**
> >
> > Thank you for the answers. I have decided to keep my score as it is.
> >
> >
> >
> > MINOR:
> >
> > My question "Why not using directly the primal formulation of KALE?" was actually:
> > Can we obtain another finite dimensional problem by starting from the primal form? Something like (7) but different from (7). Or, do we end up with (7) anyway?

---

> > > ### Author Response · Authors · 2021-08-25
> > > **Answer about KALE's primal problem**
> > >
> > > Thank you for your comment.
> > >
> > > To answer your question, indeed we can obtain a primal, finite-dimensional analogue of the problem in Eq. 2, which also defines the KALE. This primal problem is obtained using the Representer Theorem, and has a different form compared to the (dual) problem given in Eq. 7.
> > >
> > > However, relying on Fenchel duality, one can show a 1 to 1 correspondence between the primal solution $h^\star$, and $f^\star$, a function characterizing the dual solution. This correspondence is given for the population case in the second line of Eq. 6, and reduces to a finite sum for the sample-based setting. Again from Fenchel duality, we get that at the optimum, the primal and the dual problem have the same value. We will clarify this point in the paper.

---

### Official Review · Reviewer_WiM9 · 2021-07-15

**Rating:** 6
**Confidence:** 4

**Summary:**

The authors propose KALE Flow to transport mass from one distribution to another. They also study the properties and convergence of the proposed algorithm.

**Ethical Concerns:**

No.

**Limitations And Societal Impact:**

No.

**Main Review:**


This paper provides a good theoretical analysis of KALE, and introduces a particle descent algorithm.
However,  as the authors stated, Arbel M, Zhou L, Gretton A.  2021 studied KALE previously.  The main contribution in this paper  compared with Arbel M, Zhou L, Gretton A.  2021, is   the optimization guarantees. However, the main idear of this  part is similar to the MMD flow paper. For the numerics,  I am not really convinced by the toy experiments.

 It is better to present several concrete examples, in which the kernelized KL divergence has a closed-form solution. Or the authors may provide several toy examples to compare the advantage of the proposed generalized KL divergences.
Some  related works are missing, for example,  Li et.al, Natural gradient via optimal transport, Information geometry, 2018; Li, Hessian metric via transport information geometry, 2020.

Another issue is  the exponential dependence on the iteration number $n$ seems unavoidable in the current context. If my understanding is right, KALE flow converges at an extremely slow rate $O(poly(1/\log n))$ by choosing $n = O(\log N)$.
Therefore,  the KALE flow becomes less appealing.





**Time Spent Reviewing:**

5

---

> ### Author Response · Authors · 2021-08-10
> **Thank you for your review**
>
> We thank Reviewer WiM9 for their suggestions, and respond to specific points below:
>
> *The main contribution in this paper compared with Arbel M, Zhou L, Gretton A. 2021, is the optimization guarantees. However, the main idea of this part is similar to the MMD flow paper.*
>
> We respectfully disagree with this remark:
>
> The work built around KALE in [5] considers a parametric, stochastic optimization of the KALE. Our setting is fully nonparametric and uses a particle algorithm, and is thus different from the setting considered in the prior work. Moreover, we believe that the contributions of this paper are not limited to the "optimization guarantees" of the KALE flow (which we assume refer to Proposition 3 and Proposition 5). Indeed, unrelated to the these optimization guarantees:
>
> - We build on the definition of the KALE in Arbel, Zhou, Gretton to propose another KALE definition that is adapted for the gradient flow setting. We build on proof techniques in  Arbel, Zhou, Gretton to prove that the latter KALE is also a probability divergence. Importantly, we also show that  the latter KALE interpolates between the KL and MMD. This interpolation property allows us to make high-level predictions about the "geometry" of the KALE, which we confirm in the experiment section.
> - We formally prove the existence and uniqueness of the KALE gradient flow. The KALE being defined as a non closed-form optimum over a functional space, proving such results requires a considerable amount of care (pages 20 to 23).
> We welcome suggestions of the reviewers on ways to clarify the extent of our contributions in the main paper.
>
> *For the numerics, I am not really convinced by the toy experiments.*
>
> We believe that the experiments empirically confirm the following two points:
> The KALE is more sensitive to the support of the distributions than the MMD, a claim that is partly justified by the fact that the KALE approaches the KL (a divergence that forbids non-overlapping supports) as $\lambda \to 0$.
> The KALE flow trajectories are "better behaved" than the MMD flow trajectories, while remaining well defined for mutually singular probabilities, which is not the case for the KL flow. Thus, the KALE flow provides a valuable alternative to both the KL and the MMD flow for probabilities with disjoint support.
>
> *It is better to present several concrete examples, in which the kernelized KL divergence has a closed-form solution.*
>
> Closed-form examples would be indeed valuable, for instance to understand the impact of the regularization parameter $\lambda$ on the KALE and its witness function. However, KALE requires solving a QP, and we are not aware of any non-trivial triplet of probabilities/regularizer $(\mathbb P, \mathbb Q, \lambda)$ for which the solution has a closed form.
>
> *Or the authors may provide several toy examples to compare the advantage of the proposed generalized KL divergences.*
>
> We believe that the toy examples provided in Section 5 do illustrate the advantages of the KALE over the MMD and the KL: a greater sensitivity to support mismatch than the MMD leading to better behaved KALE flow trajectories (compared to the MMD flow trajectories), as well as well-posedness of the flow even if $\mathbb P$ and $\mathbb Q$ are mutually singular (unlike the KL).
>
> *Another issue is the exponential dependence on the iteration number seems unavoidable in the current context. If my understanding is right, KALE flow converges at an extremely slow rate by choosing….*
>
> Provided that the KALE trajectories satisfy a boundedness assumption, the KALE flow converges at a $1/t$ rate, with a possible discrete-time extension relying on proof techniques [3] with a rate $1/n$, $n$ being the number of iterations. This rate does not match $O(\text{poly}(1/\log n))$ mentioned in the reviewer comment, and in fact we are not certain of how it was obtained. We would welcome further clarification by the reviewer, which we will be happy to address in a final version of the paper.
>
> Re the suggested relevant works: thank you for these references - on reading them, our understanding is that they relate to the parametric case, which is not our setting. We may have missed some additional connections, however, and we would be grateful for further reviewer feedback on this point.

---

> > ### Comment · Reviewer_WiM9 · 2021-08-23
> > **Thanks for the reply.**
> >
> > Thanks the authors' reply. And most of my concerns are eliminated. And I will change my sore to 6.

---

### Official Review · Reviewer_2uTN · 2021-07-20

**Rating:** 6
**Confidence:** 3

**Summary:**

Gradient flows defined on the Wasserstein space have received particular attention from the statistical and machine learning community, since they allow to fit a target distribution starting from another source distribution. They consist in minimizing a certain functional, and the choice of this functional strongly affects its theoretical properties and empirical performance. Recent work have studied gradient flows based on specific functionals, including the maximum mean discrepancy (MMD) and the Kullback-Leibler divergence (KL). This paper focuses on these two cases and proposes a novel gradient flow method based on an approximation of KL, in order to alleviate the limitations induced by MMD and KL.

First, the authors introduce a novel divergence on the space of probability measures, which corresponds to a relaxed version of the KL and is called the "KL approximate lower-bound estimator" (KALE, Definition 1). They study some of its topological properties (Theorem 1), show that KALE interpolates between KL and a specific instance of MMD if the associated kernel is bounded (Proposition 1). Besides, KALE admits a dual formulation (equation (6)), which can be approximated using samples from the two distributions of interest (eq. (7)).

Then, the authors develop the gradient flow in the Wasserstein space based on KALE, by solving a specific instance of the continuity equation in eq. (8): see Proposition 2. The resulting gradient flow is shown to ensure global convergence under certain assumptions (Proposition 3). A practical algorithm is developed to approximate the solution of the KALE gradient flow, and imply discretizing the time of the continuity equation and performing gradient descent on finitely many samples (Algorithm 1). This "particle descent" scheme induce an approximation error (due to the use of samples instead of their underlying densities), which is upper bounded in Proposition 4. The authors also present a variant of this algorithm, largely inspired by prior work on MMD flows, which incorporate Gaussian noise (Section 4.2). The goal of this approach is to relax the convergence guarantees associated to time discretization, which are too strict (Proposition 5). Experiments on different synthetic settings are finally conducted to illustrate the advantages yielded by the KALE gradient flow over the MMD and KL counterparts (Section 5).


**Ethical Concerns:**

This paper does not seem to raise any ethical issues.


**Limitations And Societal Impact:**

The authors addressed two limitations: (i) the condition on the noise schedule in Proposition 5 might be too strict, given Figures 3 to 5; (ii) KALE flow (Figure 2) does not recover outliers well (Figure 2, l.303-305).

The societal impact is not discussed.


**Main Review:**

This work is relevant for the NeurIPS community, since prior work have demonstrated that gradient flows can be employed in modern machine learning approaches, including generative modeling. Moreover, the newly introduced probability divergence KALE is clearly motivated: according to the empirical analysis in Section 5, KALE can return useful information when comparing two distributions, even when their support do not overlap (unlike KL), and might offer a higher discriminative power than MMD. These advantages, combined with the topological properties in Theorem 1 and the fact that it can be estimated when the two distributions are accessible through samples only, suggest that KALE has great potential for other applications than gradient flows.

The paper is very dense as it provides numerous theoretical contribution on novel tools, and due to lack of time, I was not able to thoroughly check all the proofs in the appendix. The empirical analysis in Section 5 illustrates well the flexibility of KALE gradient flow over the MMD and KL ones, likely due to its interpolation property (Proposition 2). The rest of the claims were not empirically verified, and in my opinion, some of them should be discussed more:

(A1) "As shown in [5], the KALE's sample complexity scales with $\frac1{\sqrt{N}}$" (l.157-158), where [5] refers to "Generalized energy based models" (Arbel et al., 2021). However, the authors also emphasized that the KALE they consider "differs from the one in [5]" (l.111). In that sense, it is not clear why the sample complexity in [5] still applies here. On the other hand, that sample complexity is said to be "similar to the sample complexity of Sinkhorn divergences". The sample complexity of Sinkhorn divergences also strongly depends on the value of its regularization coefficient; is it the case for KALE?

(A2) The convergence rate of KALE flow derived in Proposition 3 is said to be slower than the one yielded by KL flow (l.192), but it is not clear how it compares with MMD flow (in particular, with Proposition 7 in "Maximum Mean Discrepancy Gradient Flow", Arbel et al., 2019). Therefore, I have trouble understanding why "trajectories of the KALE flow exhibit better convergence properies when compared to the MMD flow" (l.326-327): it this claim solely supported by the empirical results in Section 5 for now?

(A3) Proposition 4 establishes an approximation error bound, which guarantees that "given sufficiently many samples of $\mathbb{P}_0$ and $\mathbb{Q}$, one can approximate an exact discrete KALE flow ... with arbitrary precision" (l.240-242). However, the derived bound depends on other constants than the number of samples, whose impact is not discussed at all. In particular, the smaller $\lambda$, the larger $A$ (l.239), so the larger $N$ needs to be to compensate. This might be a limitation of KALE in practice and it should be addressed in the paper to give a more sensible interpretation.

The aforementioned points explain why I can't give a higher overall score for now.

This paper is very well written, but some parts in the main document are not self-contained, which makes them hard to understand. I encourage the authors to address the following points to improve readability.

(B1) Assumptions 1 to 3, which play an important role in most theoretical results in this work, are only presented in the appendix (Section B). The main document should at least provide a small description of them. Besides, it is not clear to which extent these assumptions are met in the experiments.

(B2) The notation $\dot{H}^{-1}(\mathbb{P}_t)$ in the negative Sobolev norm is not clear. The definition of this norm (given in Section F) could be recalled in Proposition 3.

(B3) The set $\mathcal{H}_n$ in equation (13) is not defined.

(B4) The constants in Proposition 4 ($L$, $K$, $\mu$) are not explained. This point is actually connected to (B1), since these constants come from the assumptions.

Some other minor inaccuracies and typos are given below.

l.97: "an superscript" -> "a superscript"

Equation (7): Given equation (6), why is it -1 instead of +1?

Equation (11): $\partial \mathbb{P}_t$ -> $\partial_t \mathbb{P}_t$

Equation (13): What is the norm used here? (Precise the set)

l.214 "Approxation" -> "Approximation"

Equation (15): the set of $h$ should be precised.

Algorithm 1:
- Extra "max_iter",
- The variable $\texttt{v}$ should depend on $\texttt{h}$_$\texttt{star}$,
- Precise what operation $\texttt{log}$_$\texttt{ratio}$ does.

l.257: Given the rest of the statement, $\mathcal{D}_{\beta_n}$ should be defined as a function of $\mathbb{P}_n$.

l.272: "appearing in the the"

l.284: "and thus the the"

Figure 3: in the legend, "and and"

l.624: "ahd" -> "and"

l.646: "metrices" -> "metrizes"

Equation below l.651: $\mathcal{K}(0, \mathbb{P})$ -> $\mathcal{K}(0_{\mathcal{H}}, \mathbb{P})$

Equation (27): $\mathbb{P}$ should be $\mathbb{P}_t$



**Time Spent Reviewing:**

6

---

> ### Author Response · Authors · 2021-08-10
> **Thank you for your review**
>
> We thank Reviewer 2uTN for their thorough analysis of our work. We gratefully acknowledge the proposed suggestions, which we will implement in the revised version of the paper.
>
> (B1) *Assumptions 1 to 3, which play an important role in most theoretical results in this work, are only presented in the appendix (Section B). The main document should at least provide a small description of them. Besides, it is not clear to which extent these assumptions are met in the experiments.*
>
> We agree with this comment. We stress that these assumptions are smoothness and boundedness assumptions that are met in the experiments, since the experiments use the very smooth gaussian kernel.
>
>
> *(A1) (a) "As shown in [5], the KALE's sample complexity scales with $1/\sqrt{N}$" (l.157-158), where [5] refers to "Generalized energy based models" (Arbel et al., 2021). However, the authors also emphasized that the KALE they consider "differs from the one in [5]" (l.111). In that sense, it is not clear why the sample complexity in [5] still applies here*
>
> We agree that the discussion around sample complexity was not clear. The sentence "As shown in [5], the KALE's sample complexity scales with $1/\sqrt{N}$" requires additional clarification when used in our paper, for two reasons:
> - KALE in [5] is slightly different to KALE in the present submission, as the latter retains the $\lambda  \|\| h \|\|^2_H /2$   term in its definition.
> - [5, Theorem 7] proves that the $\text{KALE}(\hat{\mathbb P}, \hat{\mathbb Q})$ converges to $\text{KL}(\mathbb P, \mathbb Q)$. This requires two steps: taking the limit of infinite samples, and considering $\lambda \rightarrow 0$. In the present paper "sample complexity" refers only to the first of the two steps: convergence of $\text{KALE}(\hat{\mathbb P}, \hat{\mathbb Q})$ to $\text{KALE}(\mathbb P, \mathbb Q)$ (with $\lambda$ fixed, and additionally $\lambda \|\| h \|\|^2_\mathcal H/2$ included).
> The adaptation from [5] to our paper is nonetheless straightforward (as you might expect), and we will include this clarification in our final version.
>
> (A1) (b) *On the other hand, that sample complexity is said to be "similar to the sample complexity of Sinkhorn divergences". The sample complexity of Sinkhorn divergences also strongly depends on the value of its regularization coefficient; is it the case for KALE?*
>
> This is a very interesting point: Using arguments similar to [5, Theorem 7], we show, under a smoothness assumption ($\log \text{d}\mathbb P/\text{d}\mathbb Q \in \mathcal H$), that the sample complexity of the KALE is in $O(1/(\lambda N) + 1/\sqrt{N})$. By contrast, [21, Theorem 3] shows that the empirical Sinkhorn divergence converges to its population counterpart as $O(e^{\kappa/\lambda}/(\sqrt N (1 + \lambda^{\lfloor{d/2} \rfloor}))$. Note that the lambdas in each case have different meanings: RKHS norm regularization for KALE; entropic regularization for Sinkhorn.
>
> Making a direct comparison of these bounds would be tricky ; however a way to compare Sinkhorn and KALE is to regard them as empirical estimators of Wasserstein and the KL: one can then study, for a fixed sample size N, the value of $\lambda$ leading to the lowest approximation error.
> - In the case of Sinkhorn divergences [A, proposition 5] shows that in the case of distributions with bounded support and densities, setting $\lambda$ to $N^{-1/d}$ leads to a precision of order $N^{-2/(d+4)}$.
> - In the case of the KALE, and under the smoothness assumption $\log \text{d}\mathbb P/\text{d}\mathbb Q \in \mathcal H$, setting $\lambda$ to  $1/\sqrt N$ leads to a $O(1/\sqrt N)$ approximation error with the KL. Compared to Sinkhorn, we see that our smoothness assumption allows us to get dimension-free approximation bounds.
>
>
>
> (A2) *The convergence rate of KALE flow derived in Proposition 3 is said to be slower than the one yielded by KL flow (l.192), but it is not clear how it compares with MMD flow (in particular, with Proposition 7 in "Maximum Mean Discrepancy Gradient Flow", Arbel et al., 2019). Therefore, I have trouble understanding why "trajectories of the KALE flow exhibit better convergence properties when compared to the MMD flow" (l.326-327): it this claim solely supported by the empirical results in Section 5 for now?*
>
> The reviewer is correct. Instead, we will replace this sentence (l.326-327) by:
> "Using the KALE Particle Descent Algorithm we have shown **on several examples** that in cases where a KL Gradient Flow cannot be defined, trajectories of the KALE flow **empirically** exhibit better convergence properties when compared to the MMD Flow, a flow that the KALE is also able to interpolate.".
>
> (A3) *Proposition 4 establishes an approximation error bound, which guarantees that "given sufficiently many samples of $\mathbb P$ and $\mathbb Q$ , one can approximate an exact discrete KALE flow ... with arbitrary precision" (l.240-242). However, the derived bound depends on other constants than the number of samples, whose impact is not discussed at all. In particular, the smaller , the larger (l.239), so the larger needs to be to compensate.This might be a limitation of KALE in practice and it should be addressed in the paper to give a more sensible interpretation.*
>
> The reviewer is correct. We agree that a discussion of the origin of the role of the constants in proposition 4 would be valuable, and will include one in the revised version of the paper. In short, propagation of chaos bounds are dependent on the smoothness of the dynamics. Here, $K$, $\lambda$ and $L$ and $\mu$ are boundedness constants, regularizers coefficients and Lipschitz constants that all affect the smoothness of the dynamics, hence their presence in the bound. The dependence on these smoothness parameters is typical in the literature on bounds of this type; see [26] for an analysis of the general case, [3, Theorem 9] for a similar bound to the one in our submission.
>
> Re the additional points to improve readability: thank you for these suggestions, we will implement them in a final version of the paper.
>
> ## References
> [A] Chizat, L., Roussillon, P., Léger, F., Vialard, F. X., & Peyré, G. (2020). Faster wasserstein distance estimation with the sinkhorn divergence. Advances in Neural Information Processing Systems, 33.

---

> > ### Comment · Reviewer_2uTN · 2021-08-31
> > **Post rebuttal**
> >
> > Thank you for your detailed response, which adequately addressed my questions. I think the discussion raised some interesting points, especially regarding the sample complexity: the dimension-free approximation bounds (A1(b)) is another nice feature of KALE. I recommend acceptance for this work.

---

### Decision · Program_Chairs · 2021-09-27

**Decision:**

Accept (Poster)

**Comment:**

This paper develops KALE particle flow, although the paper is incremental and the noise injection scheme is an adaption from MMD flow, reviewers and the AC agreed that the  theoretical contribution of the paper is reasonable  but the experimental side of the paper is rather weak. Weak accept